# Enhanced fitness of SARS-CoV-2 variant of concern Alpha but not Beta

Lorenz Ulrich[1,17], Nico Joel Halwe[1,17], Adriano Taddeo[2,3,17], Nadine Ebert[2,3,17], Jacob Schön[1], Christelle Devisme[2,3], Bettina Salome Trüeb[2,3,4], Bernd Hoffmann[1], Manon Wider[5], Xiaoyu Fan[6], Meriem Bekliz[7], Manel Essaidi-Laziosi[7], Marie Luisa Schmidt[8], Daniela Niemeyer[8,9], Victor Max Corman[8,9], Anna Kraft[1], Aurélie Godel[2,3], Laura Laloli[5,10], Jenna N. Kelly[2,3], Brenda M. Calderon[6], Angele Breithaupt[11], Claudia Wylezich[1], Inês Berenguer Veiga[2,3], Mitra Gultom[5,10], Sarah Osman[6], Bin Zhou[6], Kenneth Adea[7], Benjamin Meyer[12], Christiane S. Eberhardt[12,13,14], Lisa Thomann[2,3], Monika Gsell[5], Fabien Labroussaa[4], Jörg Jores[4], Artur Summerfield[2,3], Christian Drosten[8,9], Isabella Anne Eckerle[7,15,16], David E. Wentworth[6], Ronald Dijkman[5,18], Donata Hoffmann[1,18], Volker Thiel[2,3,18 ✉], Martin Beer[1,18 ✉] & Charaf Benarafa[2,3,18 ✉]

Emerging variants of concern (VOCs) are driving the COVID-19 pandemic[1,2]. Experimental assessments of replication and transmission of major VOCs and progenitors are needed to understand the mechanisms of replication and transmission of VOCs[3]. Here we show that the spike protein (S) from Alpha (also known as B.1.1.7) and Beta (B.1.351) VOCs had a greater affinity towards the human angiotensin-converting enzyme 2 (ACE2) receptor than that of the progenitor variant S(D614G) in vitro. Progenitor variant virus expressing S(D614G) (wt-S$^{614G}$) and the Alpha variant showed similar replication kinetics in human nasal airway epithelial cultures, whereas the Beta variant was outcompeted by both. In vivo, competition experiments showed a clear fitness advantage of Alpha over wt-S$^{614G}$ in ferrets and two mouse models—the substitutions in S were major drivers of the fitness advantage. In hamsters, which support high viral replication levels, Alpha and wt-S$^{614G}$ showed similar fitness. By contrast, Beta was outcompeted by Alpha and wt-S$^{614G}$ in hamsters and in mice expressing human ACE2. Our study highlights the importance of using multiple models to characterize fitness of VOCs and demonstrates that Alpha is adapted for replication in the upper respiratory tract and shows enhanced transmission in vivo in restrictive models, whereas Beta does not overcome Alpha or wt-S$^{614G}$ in naive animals.

Uncontrolled transmission of SARS-CoV-2 in the human population has contributed to the persistence of the COVID-19 pandemic. The emergence of new variants in largely immunologically naive populations suggests that adaptive mutations in the viral genome continue to improve the fitness of this zoonotic virus. In March 2020, a single amino acid change in the S protein at position 614 (S(D614G)) was identified in a small fraction of sequenced samples—this became the predominant variant worldwide within a few weeks[4]. The fitness advantage conferred by this single amino acid change was supported by major increases in infectivity, viral load and transmissibility in vitro and in animal models[3,5,6].

In the second half of 2020, SARS-CoV-2 VOCs with a combination of several mutations emerged, including Alpha, first described in southeast England[7], and Beta, first identified in South Africa[8]. In February–March 2021, Alpha rapidly became the prevailing variant in many regions of the world and a higher reproduction number was inferred from early epidemiological data[9–11]. Beyond S(D614G), Alpha has 18 further mutations in its genome compared with the progenitor, with two deletions and six substitutions within the *S* gene[12]. Some of the S mutations, such as N501Y and the H69/V70 deletion, have been hypothesized to enhance replication and transmission, but there is a lack of clear experimental evidence for this[13,14]. Beta has nine mutations in S, including N501Y,

[1]Institute of Diagnostic Virology, Friedrich-Loeffler-Institut, Greifswald-Insel Riems, Germany. [2]Institute of Virology and Immunology, Mittelhäusern, Switzerland. [3]Department of Infectious Diseases and Pathobiology, Vetsuisse Faculty, University of Bern, Bern, Switzerland. [4]Institute of Veterinary Bacteriology, Vetsuisse Faculty, University of Bern, Bern, Switzerland. [5]Institute for Infectious Diseases, University of Bern, Bern, Switzerland. [6]CDC COVID-19 Emergency Response, Centers for Disease Control and Prevention, Atlanta, GA, USA. [7]Department of Microbiology and Molecular Medicine, Faculty of Medicine, University of Geneva, Geneva, Switzerland. [8]Charité–Universitätsmedizin Berlin, Institute of Virology, Berlin, Germany. [9]German Centre for Infection Research (DZIF), Berlin, Germany. [10]Graduate School for Biomedical Science, University of Bern, Bern, Switzerland. [11]Department of Experimental Animal Facilities and Biorisk Management, Friedrich-Loeffler-Institut, Greifswald-Insel Riems, Germany. [12]Center for Vaccinology, Department of Pathology and Immunology, University of Geneva, Geneva, Switzerland. [13]Division of General Paediatrics, Department of Woman, Child and Adolescent Medicine, Faculty of Medicine, University of Geneva, Geneva, Switzerland. [14]Center for Vaccinology, Geneva University Hospitals, Geneva, Switzerland. [15]Division of Infectious Disease, Geneva University Hospitals, Geneva, Switzerland. [16]Division of Laboratory Medicine, Laboratory of Virology, Geneva University Hospitals, Geneva, Switzerland. [17]These authors contributed equally: Lorenz Ulrich, Nico Joel Halwe, Adriano Taddeo, Nadine Ebert. [18]These authors jointly supervised this work: Ronald Dijkman, Donata Hoffmann, Volker Thiel, Martin Beer, Charaf Benarafa. ✉e-mail: volker.thiel@vetsuisse.unibe.ch; martin.beer@fli.de; charaf.benarafa@vetsuisse.unibe.ch

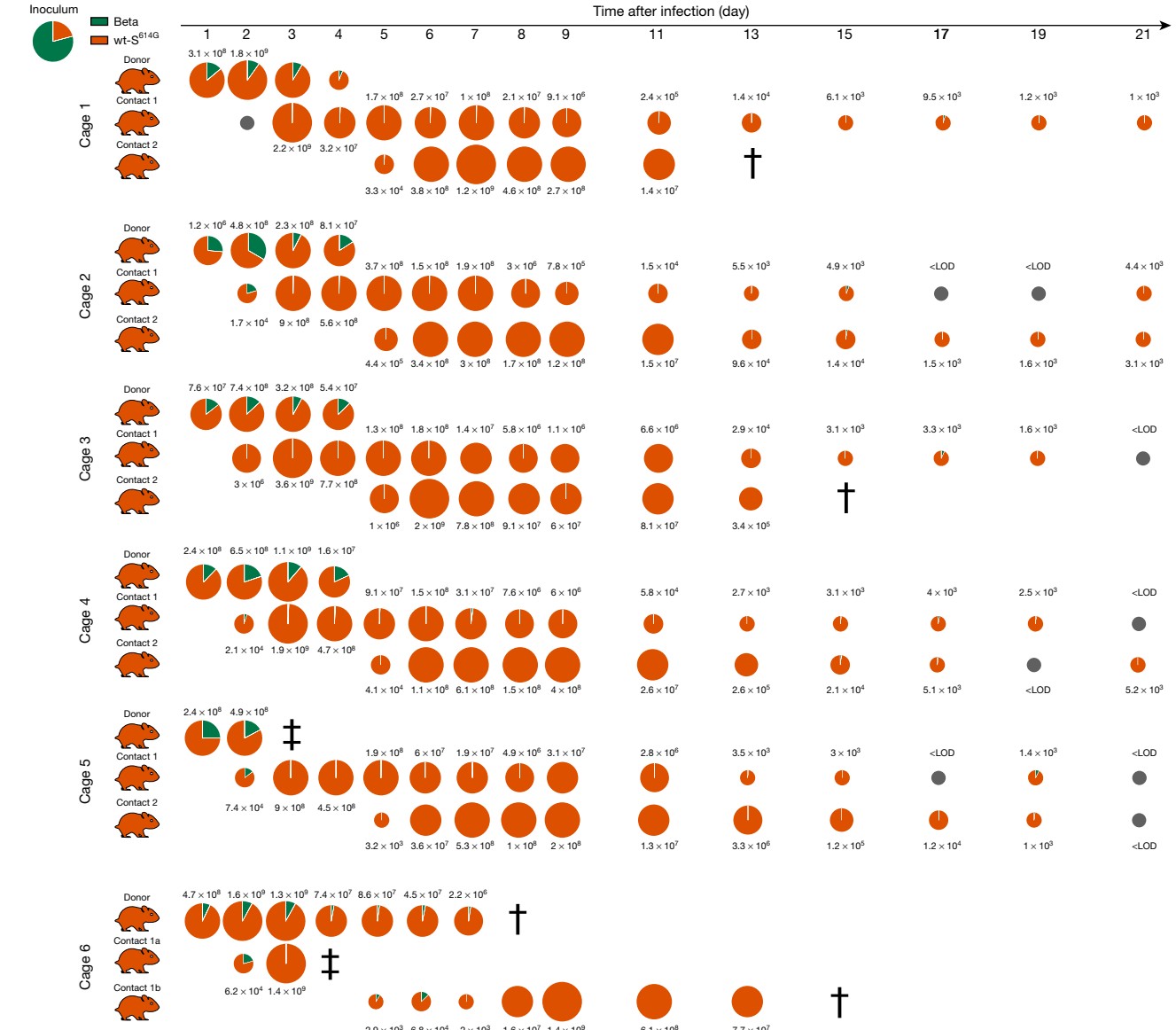

**Fig. 1 | Competitive replication and transmission of Beta and wt-S$^{614G}$ in Syrian hamsters.** Six donor hamsters were each inoculated with a median tissue culture infectious dose (TCID$_{50}$) of $10^{4.25}$, determined by back titration and comprising a mixture of wt-S$^{614G}$ (orange) and Beta (green) at a 1:3.8 ratio, determined by quantitative PCR with reverse transcription (RT–qPCR). Donor, contact 1 and contact 2 hamsters were co-housed sequentially as shown in Extended Data Fig. 2a. Nasal washes were performed daily from 1–9 dpi and then every 2 days until 21 dpi. Pie charts show the ratio of variants detected in nasal washes at the indicated dpi. Pie chart sizes are proportional to the total number of viral genome copies per ml, as shown above or below each chart. Grey pies indicate values below the limit of detection (LOD; <10$^3$ viral genome copies per ml). Hamster silhouettes are coloured according to the dominant variant (>66%) detected in the last positive sample from each animal. Daggers indicate that the animal reached the humane endpoint; double daggers indicate a hamster that died during inhalation anaesthesia at 3 and 4 dpi. This required changes in the group composition in cage 6—the donor hamster was kept until 7 dpi and was co-housed in two different pairs: donor–contact 1a and donor–contact 1b.

and two in the S receptor-binding domain (RBD), K417N and E484K. E484K is thought to be responsible for the ability of Beta to escape neutralization by plasma from convalescent individuals[15–17]. Whether S mutations are solely responsible for the putative fitness advantage and if so, which ones, remains unknown.

Here we investigate the fitness of Alpha and Beta VOCs relative to wt-S$^{614G}$, the predominant parental strain containing the S(D614G) substitution—in relevant primary airway culture systems in vitro, and in ferrets, Syrian hamsters and two mouse models expressing human ACE2—to assess specific advantages in replication and transmission and to evaluate the effects of Alpha S mutations alone in vivo. Neither Alpha nor Beta showed enhanced replication in human airway epithelial cell (AEC) cultures compared with wt-S$^{614G}$. Competitive transmission experiments in Syrian hamsters showed similar replication and transmission of wt-S$^{614G}$ and Alpha, which both outcompeted Beta. However, competitive experiments in ferrets and transgenic mice expressing human ACE2 controlled by the *KRT18* promoter (hACE2-K18Tg), which overexpress human ACE2 in epithelial cells, showed increased fitness of Alpha compared with wt-S$^{614G}$. Finally, Alpha and a recombinant clone of progenitor virus expressing the Alpha S protein (wt-S$^{Alpha}$) both outcompeted the parental wt-S$^{614G}$ strain, resulting in higher virus load in the upper respiratory tract (URT) of mice expressing human ACE2 instead of mouse ACE2 under the endogenous mouse *Ace2* promoter (hACE2-KI mice). Similar to results from AEC cultures, Beta showed lower fitness than wt-S$^{614G}$ in hACE2-KI mice. Infections with Alpha and wt-S$^{614G}$ virus resulted in similar pathologies in all the in vivo models.

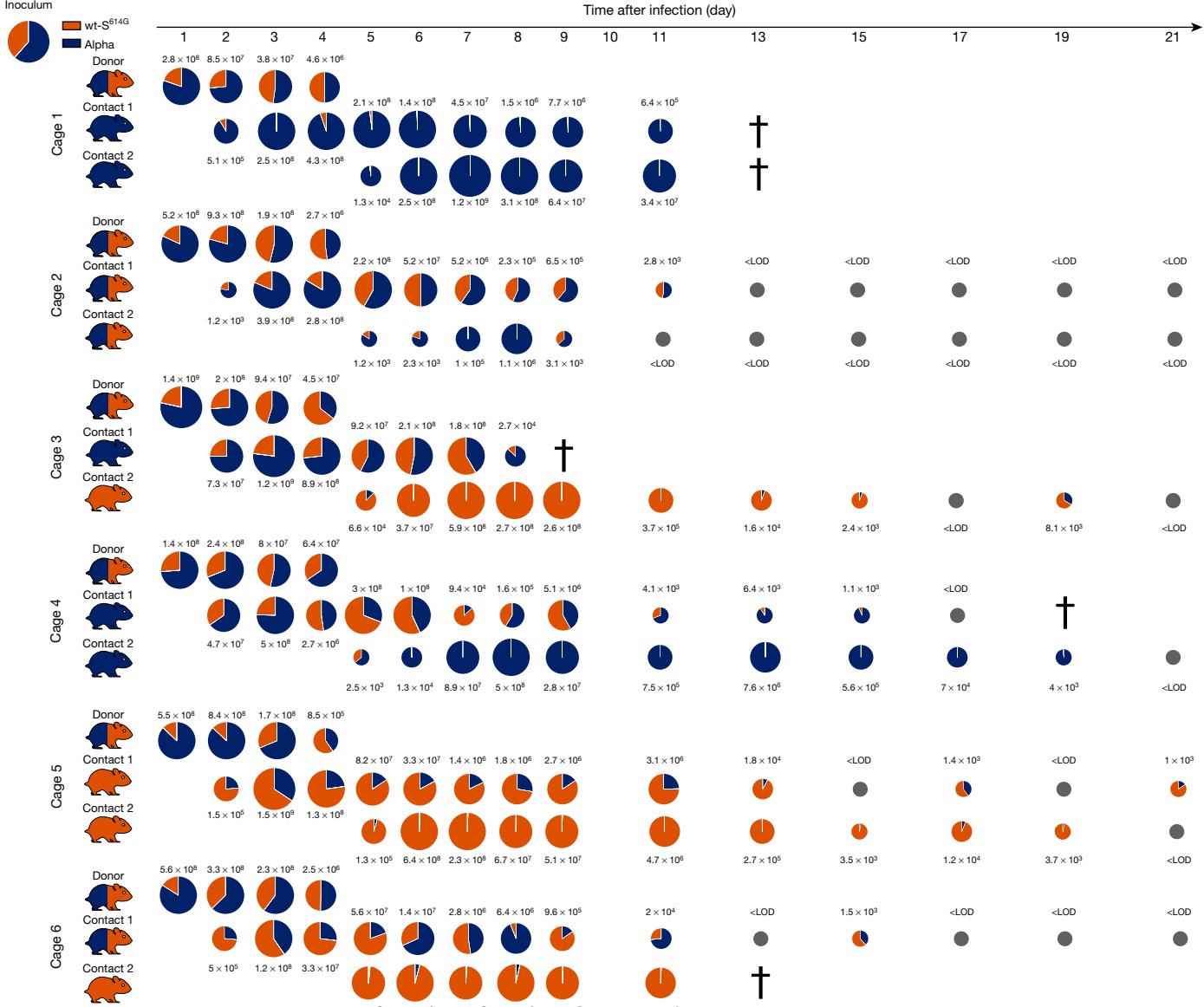

**Fig. 2 | Competitive replication and transmission of Alpha and wt-S^614G in Syrian hamsters.** Six donor hamsters were each inoculated with a TCID$_{50}$ of 10$^{4.3}$, determined by back titration and comprising a mixture of wt-S^614G and Alpha at a 1:1.6 ratio, determined by RT–qPCR. Donor, contact 1 and contact 2 hamsters were co-housed sequentially as shown in Extended Data Fig. 2a. Nasal washes were performed daily from 1–9 dpi and then every 2 days until 21 dpi. Pie charts show the ratio of variants detected in nasal washes at the indicated dpi.

Pie chart sizes are proportional to the total number of viral genome copies per ml, as shown above or below each chart. Grey pies indicate values below the LOD. Hamster silhouettes are coloured to indicate the dominant variant (>66%) detected in the last positive sample from each hamster; a silhouette with two colours indicates that there is no dominant variant. Daggers indicate that the hamster reached the humane endpoint.

## Binding and replication of VOCs in vitro

The evolution of SARS-CoV-2 variants is associated with accumulation of mutations in the S protein. We determined dissociation constant ($K_D$) values for recombinant trimeric S with immobilized dimeric human ACE2 using bio-layer interferometry. S protein from Alpha (S^Alpha) or Beta (S^Beta) exhibited a fourfold higher affinity for human ACE2 than that of S(D614G) protein (Extended Data Fig. 1a). Replication kinetics of Alpha, Beta and a wild-type clinical isolate with the S(D614G) mutation were similar in relation to viral copies and titres in AEC cultures incubated at 33 and 37 °C (Extended Data Fig. 1b). However, in direct competition experiments in AEC cultures, Alpha had no advantage over wt-S^614G, whereas Beta was outcompeted by both Alpha and wt-S^614G (Extended Data Fig. 1c), indicating that competition experiments can expose differences in replication that are not detected in individual growth kinetic assays.

## Alpha and wt-S^614G outcompete Beta in hamsters

We inoculated groups of six Syrian hamsters intranasally with a mixture of two SARS-CoV-2 strains comprising equivalent numbers of genome copies in three one-to-one competition experiments: Alpha versus Beta, Beta versus wt-S^614G, and Alpha versus wt-S^614G. All experimentally infected 'donor' hamsters were kept strictly in isolation cages to prevent intergroup spill-over infections. Each donor hamster was co-housed with a naive 'contact 1' hamster 1 day post infection (dpi), creating six donor–contact 1 pairs to evaluate shedding and transmission. At 4 dpi, donor hamsters were euthanized and six subsequent transmission pairs were set up by co-housing each contact 1 hamster with a naive contact 2 hamster (Extended Data Fig. 2a).

In two competition experiments, wt-S^614G and Alpha outcompeted Beta, as indicated by nasal washes of the donor hamsters from 1 dpi until

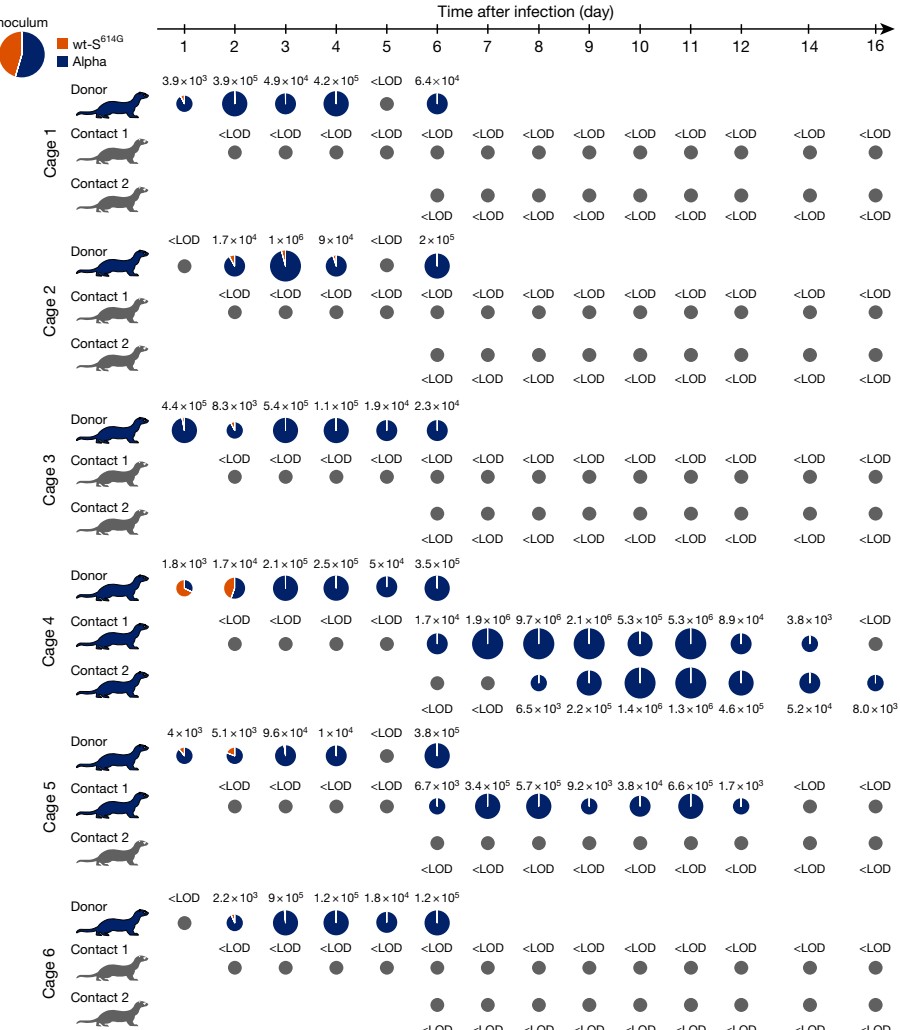

**Fig. 3 | Replication and transmission of SARS-CoV-2 Alpha and wt-S^614G in ferrets.** Six donor ferrets were each inoculated with a TCID$_{50}$ of $10^{5.9}$, determined by back titration and comprising a mixture of wt-S^614G and Alpha at a 1:1.2 ratio, determined by RT–qPCR. Donor, contact 1 and contact 2 ferrets were co-housed sequentially as shown in Extended Data Fig. 2b. Pie charts show the ratio of variants detected in nasal washes at the indicated dpi. Pie chart sizes are proportional to the total number of viral genome copies per ml, as shown above or below each chart. Grey pies indicate values below the LOD. Viral genome copies were below the LOD at 18 and 20 dpi (not shown). Ferret silhouettes are coloured to indicate the dominant SARS-CoV-2 variant (>66%) detected in the last positive sample from each ferret.

euthanasia at 4 dpi. The viral load reached $10^9$ genome copies (gc) per ml for wt-S^614G and Alpha, whereas Beta viral loads were tenfold lower at corresponding time points. Consequently, transmission of Beta was limited or undetectable in contact 1 and contact 2 hamsters compared with the competing variants wt-S^614G (Fig. 1) and Alpha (Extended Data Fig. 3). Transmission to contact hamsters was associated with clinical signs and weight loss (Extended Data Fig. 4a, b). In donor and contact hamsters, viral genome loads in the URT (comprising nasal conchae and trachea) revealed increased replication of Alpha and wt-S^614G compared with Beta (Extended Data Fig. 5a, b), which may explain the lower transmission rate of Beta in a competition context. Of note, Beta replicated to high titres in the lower respiratory tract (LRT; comprising cranial, medial and caudal lung lobes) of donor hamsters, similar levels as observed for the competing Alpha and wt-S^614G virus (Extended Data Fig. 5a, b).

Competition between Alpha and wt-S^614G showed no clear difference in virus titres in nasal washes of donor hamsters, and both variants were detected at all time points in each donor with numbers of individual variants ranging from $10^5$ to $10^9$ gc ml$^{-1}$ (Fig. 2). Of note, Alpha was dominant over wt-S^614G in the donor hamsters at 1 dpi, but these strains were balanced by the endpoint at 4 dpi. In organ samples from the donor hamsters, the highest viral loads were found in the LRT, where Alpha was predominant

(more than 66% of genome copies) overall with more than tenfold more viral genome copies than wt-S^614G in 14 out of 18 lung samples from the 6 donor hamsters (Extended Data Fig. 5c). Sequential transmission to contact animals was associated with body weight loss (Extended Data Fig. 4c) and was highly efficient for Alpha and wt-S^614G variants, which were both detected in nasal washes of almost all contact 1 hamsters (Fig. 2). Whereas all donor and contact 1 hamsters transmitted both viruses to their respective contacts, contact 2 hamsters mainly shed one variant at high levels in nasal washes, demonstrating similar transmission ability for wt-S^614G and Alpha. At the individual endpoints for contact 1 hamsters, Alpha appeared to dominate in the LRT when both variants were found at similar levels in the nasal washes and URT. In contact 2 hamsters, the variant that was dominant in the URT was also dominant in the LRT (Extended Data Fig. 5c). High levels of SARS-CoV-2 replication in hamsters induced a rapid humoral immune response, as shown by serum reactivity in RBD-based ELISA in all but one of the contact hamsters (Extended Data Fig. 6a–c). We observed a twofold increase in in vitro binding affinity of recombinant trimeric S^Alpha to hamster ACE2 compared with S(D614G) (Extended Data Fig. 1d). These findings suggest that although S^Alpha has an increased binding affinity for ACE2, this factor was not predictive of the outcome of experimental infections in hamsters.

## Alpha dominates wt-S$^{614G}$ in ferrets

In a similar approach, we inoculated six donor ferrets with a mixture of wt-S$^{614G}$ and Alpha at equivalent numbers of genome copies and monitored sequential transmission in naive contact 1 and contact 2 ferrets (Extended Data Fig. 2b). Alpha rapidly became the dominant variant in nasal washes from 2 dpi with up to 10$^5$ gc ml$^{-1}$ (Fig. 3). Correspondingly, the nasal concha of donor ferrets revealed high levels of replication in the nasal epithelium and up to 100-fold higher load of Alpha (up to 10$^{8.2}$ gc ml$^{-1}$) than wt-S$^{614G}$ (up to 10$^{6.5}$ gc ml$^{-1}$) (Extended Data Fig. 7a). Although histopathological analysis clearly indicated viral replication in the nasal epithelium of the donor ferrets (Extended Data Fig. 7b–e), we did not observe severe clinical signs of infection (Extended Data Fig. 4d, e). Transmission to contact 1 ferrets was detected in only two pairs of ferrets, and only one contact 1 ferret transmitted the virus to the contact 2 ferret. However, in each of these three transmission events, the Alpha variant was highly dominant and replicated to similarly high titres as in donor ferrets (Fig. 3). The 3 contact ferrets with virus shedding seroconverted by 15–20 days post contact (dpc), confirming active infection (Extended Data Fig. 6d).

## Alpha dominates wt-S$^{614G}$ in K18Tg mice

To assess further adaptation of Alpha to human ACE2, four hACE2-K18Tg mice, which overexpress hACE2 in respiratory epithelium[18], were inoculated with a mixture of SARS-CoV-2 wt-S$^{614G}$ and Alpha with equivalent numbers of genomic copies (Fig. 4a). Each inoculated mouse was co-housed with a contact hACE2-K18Tg mouse at 1 dpi. Alpha was dominant in the oropharyngeal samples of all four inoculated mice from 1 to 4 dpi with up to 10$^6$ gc ml$^{-1}$. The increased replicative fitness of Alpha over wt-S$^{614G}$ was further reflected throughout the respiratory tract, with higher numbers of genome copies in nose, lungs, olfactory bulb and most brain samples at 4 dpi (Fig. 4a), and inoculated mice showed loss of body weight at 4 dpi (Extended Data Fig. 8a). A relatively high infectious dose was used to promote transmission in these experiments, and was associated with high viral load (up to 10$^8$ viral genome copies per sample) in the lung and brain, leading to encephalitis—as previously reported in hACE2-K18Tg mice[19,20]. Viral loads were lower in nasal and oropharyngeal swabs from these mice, and only limited transmission was observed from these samples (two out of four contacts). None of the contact mice lost weight, but only Alpha was detectable in the lungs of contact mice at 7 dpc (Extended Data Fig. 8b).

We performed a similar competition experiment between wt-S$^{614G}$ and an isogenic recombinant virus expressing S$^{Alpha}$ (wt-S$^{Alpha}$). We inoculated hACE2-K18Tg mice with an equal mixture of wt-S$^{Alpha}$ and wt-S$^{614G}$ and housed them with a contact hACE2-K18Tg mouse at 1 dpi. The replicative advantage of wt-S$^{Alpha}$ was less clear in this experiment, and both wt-S$^{Alpha}$ and wt-S$^{614G}$ were present with similarly high numbers of viral genome copies in lung and brain samples (Fig. 4b). Transmission to contact mice was inefficient, and wt-S$^{Alpha}$ was the only virus detected in lungs of contact mice at 7 dpc (Extended Data Fig. 8b). These results indicate that the S$^{Alpha}$ spike mutations contribute to the replication advantage of Alpha over wt-S$^{614G}$ in the URT of mice that express high levels of human ACE2.

## Competition in hACE2-KI mice

To further address this question, we next used hACE2-KI homozygous mice[3]. In contrast to hACE2-K18Tg mice, hACE2-KI mice show physiological expression of human ACE2, with no ectopic expression of human ACE2 in the brain, and no expression of mouse ACE2, which has been shown to be permissive to the spike mutation N501Y contained in S$^{Alpha}$. We inoculated 4 groups of hACE2-KI mice intranasally with 10$^4$ plaque-forming units (PFU) per mouse of either wt-S$^{614G}$, Alpha, wt-S$^{Alpha}$ or Beta ($n = 8$ mice per group) as individual virus infections.

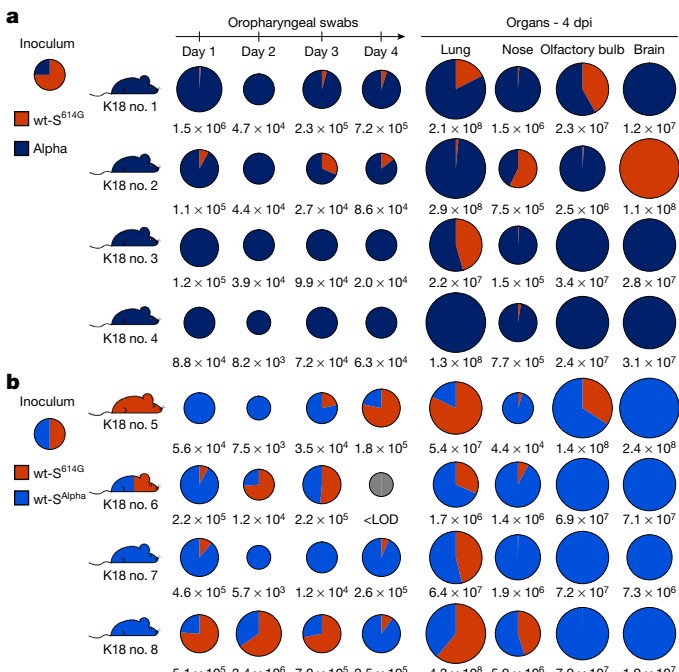

**Fig. 4 | Replication of Alpha, wt-S$^{Alpha}$, and wt-S$^{614G}$ in hACE2-K18Tg mice.**
**a**, **b**, Two groups of four donor hACE2-K18Tg mice were inoculated with 1 × 10$^4$ PFU, determined by back titration and comprising a mixture of wt-S$^{614G}$ (orange) and Alpha (dark blue) at a 3:1 ratio (**a**), or a mixture of wt-S$^{614G}$ and wt-S$^{Alpha}$ (light blue) at a 1:1 ratio (**b**). Pie charts show the ratio of variants detected in each sample at the indicated dpi. Pie chart sizes are proportional to the total number of viral genome copies per ml (swabs) or per sample (tissues), as shown below each chart. Grey pies indicate values below the LOD. Mouse silhouettes are coloured to indicate the dominant SARS-CoV-2 variant (>66%) in the last positive swab sample from the corresponding mouse; a silhouette with two colours indicates that there is no dominant variant. K18 nos. 1 to 8 denote individual hACE2-K18Tg donor mice.

We observed significantly higher viral genome copy numbers in mice infected with Alpha, wt-S$^{Alpha}$ or Beta compared with wt-S$^{614G}$ in oropharyngeal swabs at 1 dpi (Extended Data Fig. 9a). Moreover, there were significantly higher numbers of viral genome copies of Alpha and wt-S$^{Alpha}$ in the nose at 2 dpi and in the olfactory bulb at 4 dpi compared with wt-S$^{614G}$ and Beta (Extended Data Fig. 9b). Of note, viral titres in the nasal airways and lungs showed SARS-CoV-2 persistence at 4 dpi in 3 out of 4 mice infected with either Alpha or with wt-S$^{Alpha}$, but not in mice inoculated with wt-S$^{614G}$, whereas Beta persisted in the lungs of 2 out of 4 mice (Extended Data Fig. 9c). The apparent discrepancy between genome copy number and PFU reflects the non-homogeneous distribution of the virus in the different samples processed for each assay. We observed no difference in weight loss (Extended Data Fig. 9d) or lung histopathology score (Supplementary Table 1) between groups.

Finally, we performed competition experiments to compare the replication of the VOCs in groups of hACE2-KI mice. We observed a complete predominance of Alpha and wt-S$^{Alpha}$ over wt-S$^{614G}$ (Fig. 5a–c). By contrast, Beta showed reduced fitness compared with wt-S$^{614G}$ (Fig. 5d). Together, the two mouse models support enhanced fitness of SARS-CoV-2 Alpha VOC over its progenitor wt-S$^{614G}$ with increased replication and persistence in the URT and more systemic spread, mediated in part by changes in the Alpha S sequence.

## Discussion

Epidemiological data indicate that new SARS-CoV-2 variant lineages with specific amino acid changes have a fitness advantage over

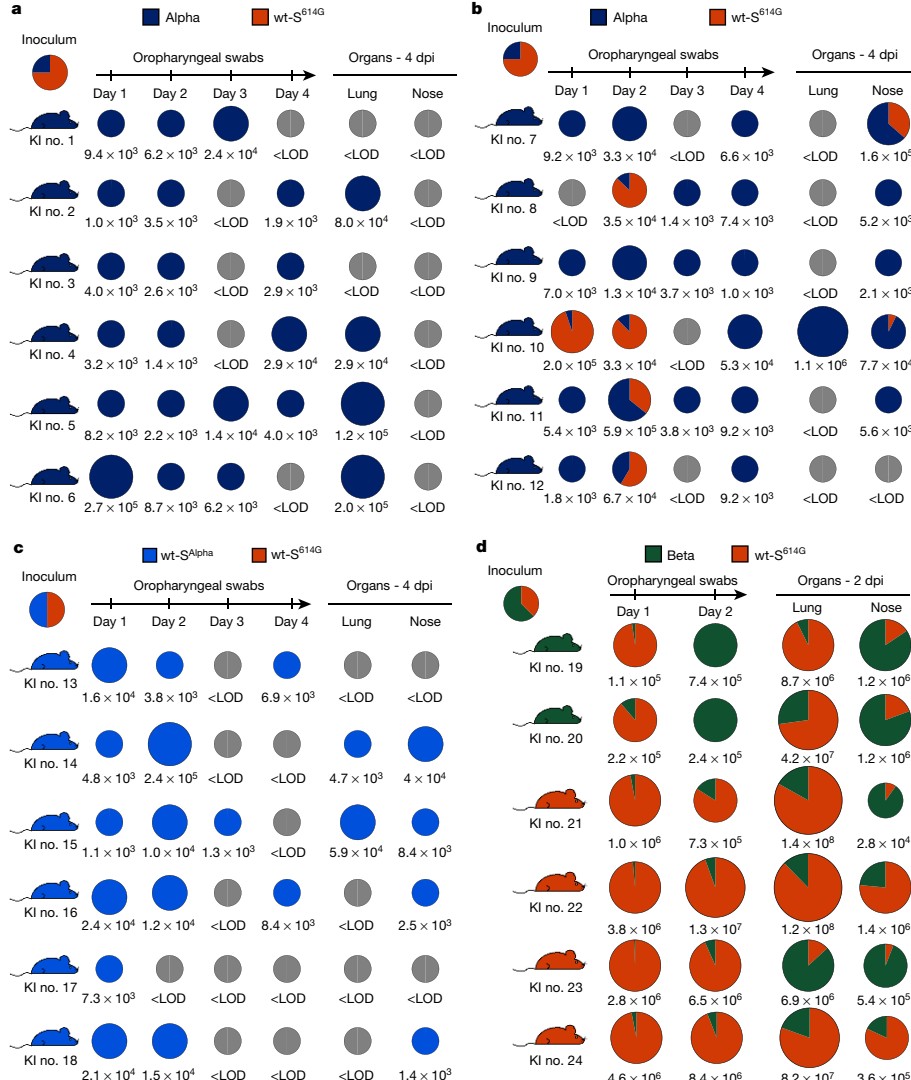

**Fig. 5 | Replication of Alpha, wt-S^Alpha, and Beta in competition with wt-S^614G in hACE2-KI mice. a–d**, Groups of hACE2-KI male (**a**, **c**, **d**) and female (**b**) mice were inoculated with $1 \times 10^4$ PFU, determined by back titration and comprising a mixture of wt-S^614G and Alpha at a 3:1 ratio (**a**, **b**), a mixture of wt-S^614G and wt-S^Alpha at a 1:1 ratio (**c**), and a mixture of wt-S^614G and Beta at a 1:1.6 ratio (**d**). Pie charts show the ratio of variants detected in each sample at the indicated dpi.

Pie chart sizes are proportional to the total number of viral genome copies per ml (swabs) or per sample (tissues), as shown below each chart. Grey pies indicate values below the LOD. Mouse silhouettes are coloured to indicate the dominant SARS-CoV-2 variant (>66%) in the last positive swab sample from the corresponding mouse. KI nos. 1 to 24 denote individual hACE2-KI mice.

contemporary strains. VOCs such as Alpha and Beta are particularly concerning for their hypothesized ability to supersede progenitor strains and establish immune escape properties, respectively. Here we provide experimental evidence that SARS-CoV-2 Alpha has a clear replication advantage over wt-S614G in ferrets and in two humanized mouse models. Moreover, Alpha was exclusively transmitted to contact animals in competition experiments, in which ferrets and hACE2-K18Tg mice were inoculated with mixtures of Alpha and wt-S^614G. Because SARS-CoV-2 replicates to lower levels in ferrets and hACE2-KI mice, the inability to detect wt-S^614G in some samples from inoculated animals also reflects the limit of detection of the assays using PCR with reverse transcription (RT–PCR) (approximately $10^3$ gc ml$^{-1}$).

We have shown that the molecular mechanism underlying the fitness advantage of Alpha in vivo is largely dependent on a few changes in S, including three amino acid deletions (H69, V70 and Y144) and six substitutions (N501Y, A570D, P681H, T716I, S982A and D1118H). In hACE2-KI mice, higher genome copies and/or titres of Alpha and wt-S^Alpha compared with wt-S^614G were found in the URT (oropharynx and nose) and olfactory bulb. Increased replication and transmission of wt-S^Alpha over

wt-S^614G were also evident in hACE2-K18Tg mice. Transmission events are rare in mice; however, we observed transmission of Alpha and wt-S^Alpha in 50% of the contact hACE2-K18Tg mice and no detection of wt-S^614G in any contact mouse. In vitro, Alpha S mutations increased its affinity for hamster and human ACE2 by twofold and fourfold, respectively, indicating an overall improvement in binding abilities rather than a specialization towards human ACE2.

Beta showed a higher binding affinity for human ACE2 than its progenitor wt-S^614G and an equal level of replication to Alpha and wt-S^614G in single infections of AEC cultures and in hACE2-KI mice. However, Beta replication was outcompeted in direct competition with wt-S^614G in vitro and in hACE2-KI mice. In hamsters, wt-S^614G and Alpha also outcompeted Beta in relation to replication and transmission to contact animals, in which Beta was outnumbered by one or two orders of magnitude. This reduced fitness was also evident in previous experiments in K18-hACE2 mice[21]. The relative reduced intrinsic fitness of Beta in immunologically naive hosts supports the hypothesis that the epidemiological advantage of Beta may be principally owing to immune escape, as indicated by reduced efficiency in serum neutralization tests[16]. In convalescent

or vaccinated populations, the immune escape advantage of Beta may prove to be sufficient to compensate its reduced intrinsic fitness and explains, for example, the low prevalence of this variant in regions with a mainly naive population.

Alpha and wt-S[614G] exhibited similar replication and transmission in hamsters, a model with very high susceptibility and replication efficacy, in which the impact of a marginally fitter SARS-CoV-2 variant may not become apparent. Indeed, efficient simultaneous transmission of both variants to contact hamsters was observed in association with high viral loads in infected animals. In models supporting high replication, such as human AEC cultures and hamsters, only major improvements in replication and transmission can be detected when the variants compared already have a high fitness. By contrast, in ferrets and mouse models—in which SARS-CoV-2 replication is overall less efficient—VOCs with modestly enhanced replication and transmission can be identified. The similar replication and transmission efficacies in hamsters are in line with recent publications using VOCs in the hamster model[22].

The basal rate of replication is an important factor in the assertion of a variant over a contemporary variant in a naive population. Some individuals with higher bioaerosol exhalation levels can initiate disproportionate numbers of transmission events, possibly because of higher viral load in the URT, and are therefore called 'superspreaders'[23]. The hamster model might thus resemble the human superspreader scenario, since there is no clear indication of a specific predominance in transmission between two SARS-CoV-2 variants with high fitness levels, such as wt-S[614G] and Alpha. However, we did not perform strict aerosol transmission studies, so this remains only a proposition. The ferret and hACE2-KI models are more restricted in that infection is predominantly in the URT. Therefore, these models more closely mimic the situation in humans, in which infections are predominantly mild. Although the rate of transmission was not high overall (3 out of 8 pairs in ferrets, and 4 out of 8 pairs in hACE2-K18Tg mice), the almost exclusive transmission of Alpha relative to wt-S[614G] mirrored increased transmission of Alpha in the human population; Alpha has been responsible for more than 90% of infections in most countries in Europe[24].

Overall, our study demonstrates that multiple complementary models are necessary to comprehensively evaluate different aspects of human SARS-CoV-2 infection and the impact of emerging VOCs on the course of the ongoing pandemic. The hamster and ferret provide complementary models for transmission efficiency. The mouse models used here may become critical for VOCs demonstrating higher specificity for binding to human ACE2 relative to those from other species. Together, our results show the clear fitness advantage of Alpha and a concomitant disadvantage of Beta, in line with the observed epidemiological predominance of Alpha in the context of a relatively naive population. Notably and reassuringly, despite the apparent fitness differences of these VOCs, there is no indication of different pathologies.

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

## Methods

### Cell lines

Vero E6 cells (ATCC CRL-1586) (provided by D. Muth, M. Müller and C. Drosten) or Vero-TMPRSS2[25] (provided by S. Pöhlmann) were propagated in Dulbecco's Modified Eagle Medium-GlutaMAX supplemented with 1 mM sodium pyruvate, 10% (v/v) heat-inactivated fetal bovine serum (FBS), 100 µg ml$^{-1}$ streptomycin, 100 IU ml$^{-1}$ penicillin, 1% (w/v) nonessential amino acids and 15 mM HEPES (Gibco). Cells were maintained at 37 °C in a humidified incubator with 5% $CO_2$.

### Viruses

Viruses are listed in Extended Data Table 1 together with the corresponding in vitro and in vivo experiments in which they were used. Specific amino acid changes are shown schematically in Extended Data Fig. 10. Contemporary clinical isolates from the B.1.160 (S$^{D614G}$) (EPI_ISL_414019), Alpha (EPI_ISL_2131446, EPI_ISL_751799 (L4549)) and Beta (EPI_ISL_803957 (L4550)) were isolated and minimally passaged on Vero E6 cells. Beta (EPI_ISL_981782) was initially isolated on A549 cells expressing human ACE2 before passaging on Vero E6 cells. SARS-CoV-2 Alpha (L4549) and Beta (L4550)[21] were received from the Robert-Koch-Institut Berlin, Germany. Isogenic variants with the Alpha spike (wt-S$^{Alpha}$) or individual Alpha spike mutations were introduced into a wild-type SARS-COV-2 'Wuhan' backbone strain comprising the D614G amino acid change (wt-S$^{614G}$), as described[3,26]. Isogenic viruses were grown on Vero-TMPRSS2 cells after one passage on human bronchial airway epithelial cells. All viruses were verified by performing whole-genome next generation sequencing (NGS). For SARS-CoV-2 Alpha (L4549, SARS-CoV-2 B.1.1.7 NW-RKI-I-0026/2020 passage 3), one silent mutation in the ORF1a (sequence position 11741) was determined (C to T with 27% T, 57% strand bias). For SARS-CoV-2 Beta (L4550, available under ENA study accession number MZ433432), one nucleotide exchange was detected (A12022C) resulting in the amino acid exchange D3923A in ORF1a and one SNP at sequence position 11730 (C to T with 41%, stand bias 52%).

For all in vivo virus competition experiments, we generated inoculum mixtures aiming for a 1:1 ratio of each variant based on virus stock titres. The reported mixture inoculum titres are based on back-titration of the inoculum mixtures and the indicated ratio of each variant was determined by standard RT–qPCR. SARS-CoV-2 wt-S$^{614G}$ (PRJEB45736; wt-S614G ID#49 vial 2) and SARS-CoV-2 Beta (L4550) were used to inoculate hamsters in the wt-S$^{614G}$ versus Beta study; SARS-CoV-2 Alpha (L4549), and SARS-CoV-2 Beta (L4550) were used for inoculation in the Alpha versus Beta hamster study. SARS-CoV-2 wt-S$^{614G}$, wt-S$^{Alpha}$, Alpha (L4549) and Beta (L4550) were used to inoculate hACE2 humanized mice in all single virus or mixed virus competition experiments.

### Next-generation sequencing

NGS was used to verify the sequence of isolates and isogenic clones prior to experimentation. RNA was extracted using the RNAdvance Tissue kit (Beckman Coulter) and the KingFisher Flex System (Thermo Fisher Scientific). Subsequently, RNA was transcribed into cDNA and sequencing libraries were generated as described[27] and were sequenced using the Ion Torrent S5XL Instrument (ThermoFisher). Samples with $C_t$ values >20 for SARS-CoV-2 were additionally treated with RNA baits (myBaits, Arbor Biosciences) for SARS-CoV-2 enrichment before sequencing[28]. Sequence datasets were analysed by reference mapping with the Genome Sequencer Software Suite (version 2.6, Roche), default software settings for quality filtering and mapping using EPI_ISL_414019 (Alpha), EPI_ISL_2131446 (Alpha) and EPI_ISL_981782 (Beta) as references. To identify potential single nucleotide polymorphisms in the read data, the variant analysis tool integrated in Geneious Prime (2019.2.3) was applied (default settings).

### AEC cultures

Human nasal AEC cultures were purchased from Epithelix (EP02MP Nasal MucilAir, pool of 14 donors). Maintenance of primary nasal AEC cultures were performed according to manufacturer's guidelines. Individual SARS-CoV-2 infections with contemporary virus isolates were conducted at either 33 °C or 37 °C as described elsewhere[29] using a multiplicity of infection (MOI) of 0.02, whereas all competition experiments and replication kinetics were performed with an MOI of 0.005 as described[30]. Quantification of viral load of individual SARS-CoV-2 infections with contemporary virus isolates was performed using the NucliSens easyMAG (BioMérieux) and RT–qPCR targeting the $E$ gene of SARS-CoV-2 as described[31,32]. In competition experiments, nucleic acids were extracted using the Quick-RNA Viral 96 kit (Zymo research) and RT–qPCR primers and probe sequences are shown in Extended Data Table 2. The viral replication of individual isogenic variants was monitored by plaque assay.

### Plaque titration assay

Viruses released into the apical compartments were titrated by plaque assay on Vero E6 cells as described[30,33]. In brief, $2 \times 10^5$ cells per ml were seeded in 24-well plates 1 day prior to titration and inoculated with tenfold serial dilutions of virus solutions. Inocula were removed 1 h post-infection and replaced with overlay medium consisting of DMEM supplemented with 1.2% Avicel (RC-581, FMC biopolymer), 15 mM HEPES, 5 or 10% heat-inactivated FBS, 100 µg ml$^{-1}$ streptomycin and 100 IU ml$^{-1}$ penicillin. Cells were incubated at 37 °C, 5% $CO_2$ for 48 h, fixed with 4% (v/v) neutral buffered formalin, and stained with crystal violet.

### Protein expression, purification and bio-layer interferometry assay

SARS-CoV-2 S protein expression plasmids were constructed to encode the ectodomain of S protein S(D614G) or S$^{Alpha}$ (residues 1–1208, with a mutated furin cleavage site and K986P/V987P substitutions) followed by a T4 fold on the trimerization domain and a polyhistidine purification tag. ACE2 protein (human, hamster or ferret) expression plasmids were constructed to encode the ectodomain of ACE2 followed by a human IgG1 Fc purification tag. The recombinant proteins were expressed using the Expi293 Expression system (ThermoFisher Scientific) and purified with HisTrap FF columns (for polyhistidine-tagged spike proteins) or with HiTrap Protein A column (for Fc-tagged ACE2 proteins) in FPLC (Cytiva) system. Recombinant proteins were further purified with Superose 6 Increase 10/300 GL column (Cytiva) as needed.

Binding affinity between the trimeric spike and dimeric ACE2 was evaluated using an Octet RED96e instrument at 30 °C with a shaking speed of 1,000 rpm (ForteBio). Anti-human IgG Fc biosensors (ForteBio) were used. Following 20 min of pre-hydration of anti-human IgG Fc biosensors and 1 min of sensor check, 7.5 nM of human ACE2–Fc, 7.5 nM of hamster ACE2–Fc in 10× kinetic buffer (ForteBio) were loaded onto the surface of anti-human IgG Fc biosensors for 5 min. After 1.5 min of baseline equilibration, 5 min of association was conducted at 10–100 nM S(D614G), S$^{Alpha}$ or S$^{Beta}$, followed by 5 min of dissociation in the same buffer, which was used for baseline equilibration. The data were collected using ForteBio Data Acquisition Software 12.0.1 and corrected by subtracting signal from the reference sample and a 1:1 binding model with global fit was used for determination of affinity constants.

### Animal experiment ethics declarations

All ferret and hamster experiments were evaluated by the responsible ethics committee of the State Office of Agriculture, Food Safety, and Fishery in Mecklenburg–Western Pomerania (LALLF M-V) and gained governmental approval under registration number LVL MV TSD/7221.3-1-004/21. Mouse studies were approved by the Commission for Animal Experimentation of the Cantonal Veterinary Office of Bern and

conducted in compliance with the Swiss Animal Welfare legislation and under license BE-43/20.

## Hamster studies

Six Syrian hamsters (*Mesocricetus auratus*) (Janvier Labs) were inoculated intranasally under a brief inhalation anaesthesia with a 70 μl mixture of two SARS-CoV-2 VOCs (wt-S$^{614G}$ and Alpha mixture, wt-S$^{614G}$ and Beta mixture, or Alpha and Beta mixture). Each inoculum was back-titrated and ratios of each variant were determined by RT–qPCR. The wt-S$^{614G}$ and Alpha mixture held a 1:1.6 ratio with TCID$_{50}$ of $10^{4.3}$ per hamster, the wt-S$^{614G}$ versus Beta mixture held a 1:3.8 ratio with TCID$_{50}$ of $10^{4.25}$ per hamster, and the Alpha versus Beta mixture held a 1.8:1 ratio with TCID$_{50}$ of $10^{5.06}$ per hamster.

Inoculated donor hamsters were isolated in individually ventilated cages for 24 h. Thereafter, contact hamster 1 was co-housed with each donor, creating six donor–contact 1 pairs (Extended Data Fig. 2a). The housing of each hamster pair was strictly separated in individual cage systems to prevent spillover between different pairs. At 4 dpi, the individual donor hamsters (inoculated animal) were euthanized. To simulate a second transmission cycle, the original contact hamsters (referred to as contact 1) were commingled with a further six naive hamsters (referred to as contact 2), which equates to another six contact 1 and contact 2 pairs (Extended Data Fig. 2a). These pairs were co-housed until the end of the study at 21 dpi. Because the first contact hamster (cage 6) in the competition trial wt-S$^{614G}$ versus Alpha, died at 2 dpc, the second contact hamster for this cage was also co-housed with the donor hamster; thus the first and second contact hamsters in this cage were labelled contact 1a and contact 1b, respectively. To enable sufficient contact between the donor hamster and contact 1b hamster, which was commingled routinely on 4 dpi, the donor hamster was euthanized at 7 dpi (instead of at 4 dpi), when it reached the humane end-point criterion for bodyweight (below 80% of 0 dpi body weight).

Viral shedding was monitored by nasal washes in addition to a daily physical examination and body weighing routine. Nasal wash samples were obtained under a short-term isoflurane anaesthesia from individual hamsters by administering 200 μl PBS to each nostril and collecting the reflux. Animals were sampled daily from 1 dpi to 9 dpi, and then every other day until 21 dpi. Under euthanasia, serum samples and an organ panel comprising representative URT and LRT tissues were collected from each hamster. All animals were observed daily for signs of clinical disease and weight loss. Hamsters reaching the humane endpoint, that is, falling below 80% of the initial body weight relative to 0 dpi, were humanely euthanized.

## Ferret studies

Similar to the hamster study, 12 ferrets (six donor ferrets and six transmission 1 ferrets) from the FLI in-house breeding were housed pairwise in strictly separated cages to prevent spillover contamination. Of these, six ferrets were inoculated with an equal 250 μl mixture of SARS-CoV-2 wt-S$^{614G}$ and Alpha. The inoculum was back-titrated and the ratio of each variant was determined by RT–qPCR. The wt-S$^{614G}$ versus Alpha mixture held a 1:1.2 ratio with $10^{5.875}$ TCID$_{50}$ distributed equally into each nostril of donor ferrets. Ferrets were separated for the first 24 h following inoculation. Subsequently, the ferret pairs were co-housed again, allowing direct contact of donor to contact 1 ferrets. All ferrets were sampled via nasal washes with 750 μl PBS per nostril under a short-term inhalation anaesthesia. Donor ferrets were sampled until euthanasia at 6 dpi, which was followed by the introduction of one additional naive contact 2 ferret per cage (*n* = 6), resulting in a 1:1 pairwise setup with contact 1 and contact 2 ferrets (Extended Data Fig. 2b). All ferrets, which were in the study group on the respective days, were sampled on the indicated days. Bodyweight, temperature and physical condition of all ferrets were monitored daily throughout the experiment. URT and LRT organ samples, as well as blood samples of all ferrets were taken at respective euthanasia time points.

Full autopsy was performed on all animals under BSL3 conditions. The lung, trachea and nasal conchae were collected and fixed in 10% neutral-buffered formalin for 21 days. The nasal atrium, decalcified nasal turbinates (cross-sections every 3–5 mm), trachea and all lung lobes were trimmed for paraffin embedding. Based on PCR results, tissue sections (3 μm) of all donors (day 6) and one recipient (no. 8, day 20) were cut and stained with haematoxylin and eosin for light microscopical examination. Immunohistochemistry was performed using an anti-SARS nucleocapsid antibody (Novus Biologicals NB100-56576, dilution 1:200) according to standardized avidin–biotin–peroxidase complex-method producing a red labelling and haematoxylin counterstain. For each immunohistochemistry staining, positive control slides and a negative control for the primary antibodies were included. Histopathology was performed on at least five consecutive tissue samples per animal, yielding comparable results in all cases. Lung tissue pathology was evaluated according to a detailed score sheet developed by Angele Breithaupt (DipECVP) (Supplementary Table 2). Evaluation and interpretation was performed by board-certified veterinary pathologists (DiplECVP) (AB, IBV).

## Mouse studies

hACE2-KI mice (B6.Cg-*Ace2*$^{tm1(ACE2)Dwnt}$) and hACE2-K18Tg mice (Tg(K18-hACE2)2Prlmn) were described previously[3,18]. All mice were produced at the specific-pathogen-free facility of the Institute of Virology and Immunology (Mittelhäusern), where they were maintained in individually ventilated cages (blue line, Tecniplast), with 12-h:12-h light:dark cycle, 22 ± 1 °C ambient temperature and 50 ± 5% humidity, autoclaved food and acidified water. At least 7 days before infection, mice were placed in individually HEPA-filtered cages (IsoCage N, Tecniplast). Mice (10 to 12 weeks old) were anaesthetized with isoflurane and infected intranasally with 20 μl per nostril with the virus inoculum described in the results section. One day after inoculation, infected hACE2-K18Tg mice were placed in the cage of another hACE2-K18Tg contact mouse. Mice were monitored daily for bodyweight loss and clinical signs. Oropharyngeal swabs were collected under brief isoflurane anaesthesia using ultrafine sterile flock swabs (Hydraflock, Puritan, 25-3318-H). The tips of the swabs were placed in 0.5 ml of RA1 lysis buffer (Macherey-Nagel, 740961) supplemented with 1% β-mercaptoethanol and vortexed. At 2 or 4 dpi, mice were euthanized, and organs were aseptically dissected. Systematic tissue sampling was performed as detailed previously[3].

## Animal specimens work up, viral RNA detection and quantification

Organ samples from ferrets and hamsters were homogenized in a 1 ml mixture composed of equal volumes of Hank's balanced salts MEM and Earle's balanced salts MEM containing 2 mM L-glutamine, 850 mg l$^{-1}$ NaHCO3, 120 mg l$^{-1}$ sodium pyruvate and 1% penicillin–streptomycin) at 300 Hz for 2 min using a Tissuelyser II (Qiagen) and centrifuged to clarify the supernatant. Organ samples from mice were either homogenized in 0.5 ml of RA1 lysis buffer supplemented with 1% β-mercaptoethanol using a Bullet Blender Tissue Homogenizer (Next-Advance) or in Tube M (Miltenyi Biotech, 130-096-335) containing 1 ml of DMEM using a gentleMACS Tissue Dissociator (Miltenyi Biotech). Nucleic acid was extracted from 100 μl of the nasal washes or 200 μl mouse oropharyngeal swabs after a short centrifugation step or 100 μl of organ sample supernatant using the NucleoMag Vet kit (Macherey Nagel). Nasal washes, oropharyngeal swabs, and organ samples were tested by RT–qPCR analysis for the ratio of the two different viruses used for inoculation, by applying two different assays, each of them specific for one variant: either the wt-S$^{614G}$, Alpha or Beta variant (Extended Data Tables 2, 3). Viral RNA copies in swabs and organs in studies using a single variant inoculum in mice were determined using the E protein RT–qPCR exactly as described[3].

Four specific RT–qPCR assays for SARS-CoV-2 wt-S$^{614G}$, Alpha and Beta were designed based on the specific genome deletions within ORF1

and the *S* gene (Extended Data Table 2). Here, virus-specific primers were used to achieve a high analytical sensitivity (less than 10 genome copies per μl template) of the PCR assays, and in samples with a high genome load of the non-matching virus.

The RT–qPCR reaction was prepared using the qScript XLT One-Step RT–qPCR ToughMix (QuantaBio) (hamsters and ferrets) or the AgPath-ID One-Step RT–PCR (ThermoFisher Scientific) (hACE2-K18Tg and hACE2-KI mice) in a volume of 12.5 μl including 1 μl of the respective FAM mix and 2.5 μl of extracted RNA. The reaction was performed for 10 min at 50 °C for reverse transcription, 1 min at 95 °C for activation, and 42 cycles of 10 s at 95 °C for denaturation, 10 s at 60 °C for annealing and 20 s at 68 °C for elongation. Fluorescence was measured during the annealing phase. RT–qPCRs were performed on a BioRad real-time CFX96 detection system (Bio-Rad) (hamsters and ferrets) or an Applied Biosystems 7500 Real-Time PCR System (ThermoFisher Scientific) (mice). Validation work was performed by comparison with established protocols (https://www.who.int/docs/default-source/coronavirus/eal-time-rt-pcr-assays-for-the-detection-of-sars-cov-2-institut-pasteur-paris.pdf?sfvrsn=3662fcb6_2 and ref. [31]).

## Serological tests of hamsters and ferrets

Serum samples from the wt-$S^{614G}$ versus Alpha, wt-$S^{614G}$ versus Beta, and Alpha versus Beta co-inoculated hamsters and ferrets were tested by ELISA for sero-reactivity against the RBD domain[34] using a Tecan i-control 2014 1.11 plate reader and data was analysed using Microsoft Excel 16.0. All samples were generated at the time point of euthanasia of the individual animal.

## Statistical analysis

Statistical analysis was performed using GraphPad Prism 8 or R[35] (version 4.1), using the packages tidyverse[36] (v1.3.1), ggpubr (v0.4.0) and rstatix (v.0.7.0). Unless noted otherwise, the results are expressed as mean ± s.d. Two-way analysis of variance (ANOVA) with Tukey honest significance differences post hoc test was used to compare competition results at different time points after infection in vitro. One-way ANOVA with Tukey's multiple comparisons test was used to compare viral genome copies or titres at different time points post infection in individual virus mouse infection studies. Significance was defined as $P < 0.05$.

## Reporting summary

Further information on research design is available in the Nature Research Reporting Summary linked to this paper.

## Data availability

Sequence data are available on the NCBI Sequence Read Archive (SRA) under the accession numbers PRJEB45736 and PRJNA784099, or in GenBank under the accession numbers MT108784, MZ433432, OL675863, OL689430 and OL689583 as shown in Extended Data Table 1. Source data are provided with this paper.

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

**Acknowledgements** We thank F. Klipp, D. Fiedler, C. Lipinski, S. Kiepert, K. Sliz and Daniel Brechbühl for animal care; M. Lange, C. Korthase, P. Zitzow, S. Schuparis, G. Cadau, P. Valenti, B. Lemaitre, C. Tougne, P. Fontannaz, P. Sattonnet and C. Alvarez for technical assistance; and M. Schmolke, B. Mazel-Sanchez and F. Abdul for providing A549-hACE2 cells. This work was supported by grants from the Swiss National Science Foundation (SNSF), grants no. 31CA30_196062 (C.B. and R.D.), 31CA30_196644 (V.T., I.A.E. and R.D.), 310030_173085 (V.T.), 310030_179260 (R.D.), 196383 (I.A.E.); the European Commission, Marie Skłodowska-Curie Innovative Training Network 'HONOURS', grant agreement no. 721367 (V.T. and R.D.); the European Union Project ReCoVer, grant no. GA101003589 (C. Drosten); Core funds of the University of Bern (V.T. and R.D.); Core funds of the German Federal Ministry of Food and Agriculture (M. Beer); the Deutsche Forschungsgemeinschaft (DFG), project no. 453012513 (M. Beer); the Horizon 2020 project 'VEO', grant agreement no. 874735 (M. Beer); COVID-19 special funds from the Swiss Federal Office of Public Health and the Swiss Federal Office of Food Safety and Veterinary Affairs (A.S. and V.T.); the Fondation Ancrage Bienfaisance du Groupe Pictet (I.A.E.); the Fondation Privée des Hoˆpitaux Universitaires de Genève (I.A.E.); and the German Ministry of Research, VARIPath, grant no. 01KI2021 (V.M.C.).

**Author contributions** Conceptualization: D.H., M. Beer, V.T. and C.B. Data curation: L.U., N.J.H., A.T., N.E., J.S., C. Devisme, B.Z. and R.D. Funding acquisition: A.S., I.A.E., D.E.W., R.D., V.T., M. Beer and C.B. Investigation: L.U., N.J.H., A.T., N.E., J.S., C. Devisme, B.S.T., B.H., M.W., X.F., M. Bekliz, M.E.-L., M.L.S., D.N., V.M.C., A.K., A.G., L.L., J.N.K., B.M.C., A.B., C.W., I.B.V., M. Gultom, S.O., B.Z., K.A., B.M., C.S.E., L.T., M. Gsell, R.D. and D.H. Methodology: B.H., A.B., F.L. and J.J. Supervision: I.A.E., C. Drosten, R.D., D.H., V.T., M. Beer and C.B. Visualization: L.U., N.J.H., A.T., J.S., C. Devisme and R.D. Writing, original draft: L.U., N.J.H., A.T., R.D., D.H., M. Beer and C.B. Writing, review and editing: L.U., N.J.H., A.B., L.T., R.D., D.H., V.T., M. Beer and C.B.

**Competing interests** The authors declare no competing interests.

**Additional information**
**Correspondence and requests for materials** should be addressed to Volker Thiel, Martin Beer or Charaf Benarafa.

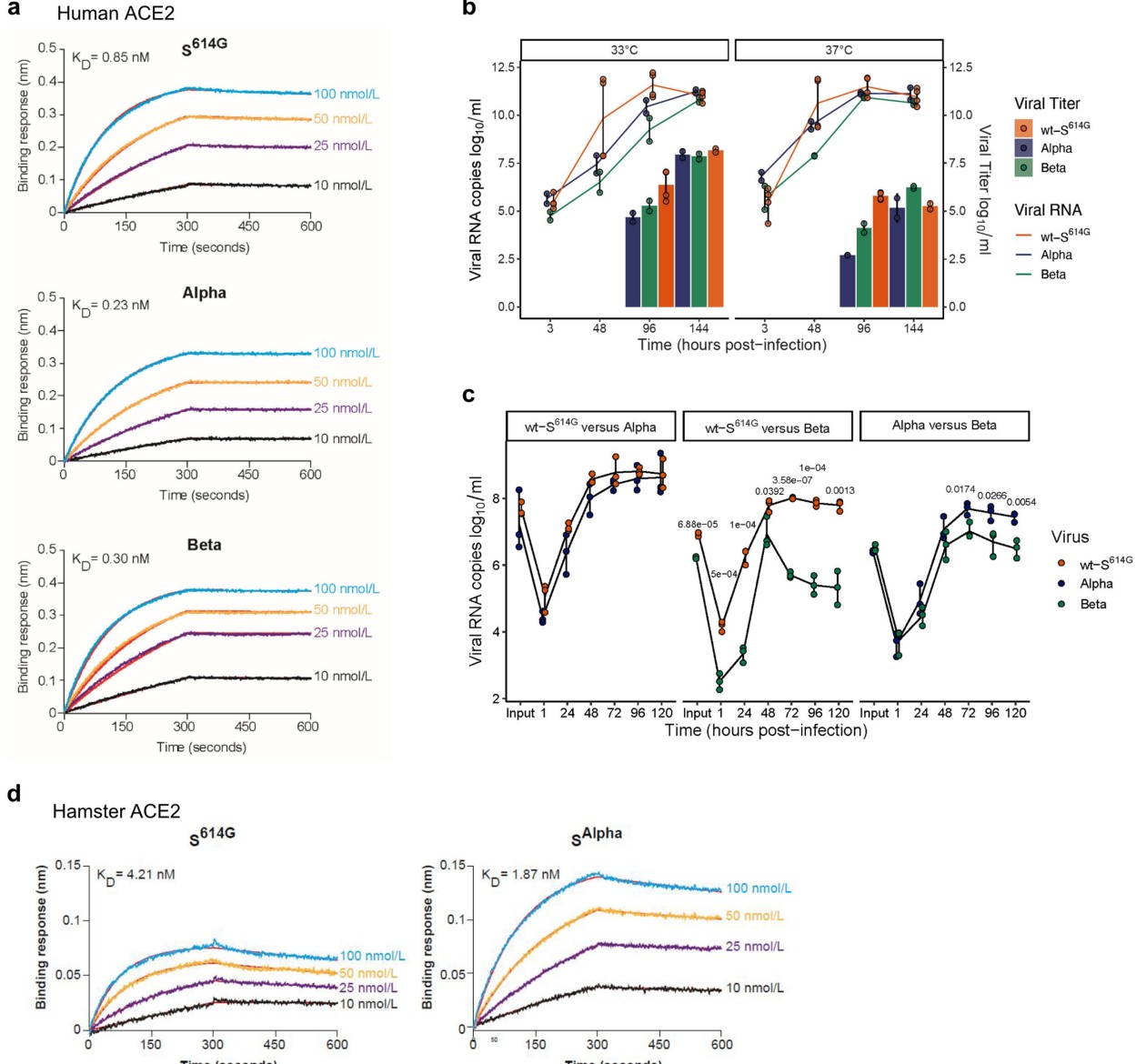

**Extended Data Fig. 1 | ACE2 receptor binding and replication kinetics of SARS-CoV-2 VOC in vitro.** (**a**) Affinity between spike (S$^{614G}$, S$^{Alpha}$, and S$^{Beta}$) protein trimers and hACE2 dimers determined by Bio-layer interferometry. (**b**) Viral replication kinetics of SARS-CoV-2 Alpha, Beta, and wt-S$^{614G}$ (MOI 0.02) at 33 °C and 37 °C in primary human nasal airway epithelial cell (AEC) cultures. (**c**) Viral replication kinetics of pairwise competition assays in primary nasal AEC cultures at 33 °C (MOI 0.005). (**b**, **c**) Data are presented as individual points with mean (line) and standard deviation; (**b**) n = 2 (Alpha and Beta), n = 4

(wt-S$^{614G}$), (**c**) n = 3 independent biological replicates. (**c**) P-values were determined by two-way ANOVA and Tukey Honest Significant Differences (HSD) post-hoc test. (**d**) Affinity between spike (S$^{614G}$, S$^{Alpha}$) protein trimers with hamster ACE2 determined by Bio-layer interferometry. (**a**, **d**) ACE2 with IgG1 Fc tag were loaded on anti-human IgG Fc biosensors and binding kinetics were conducted using indicated concentrations of spike trimers. Data is representative of 3 independent experiments.

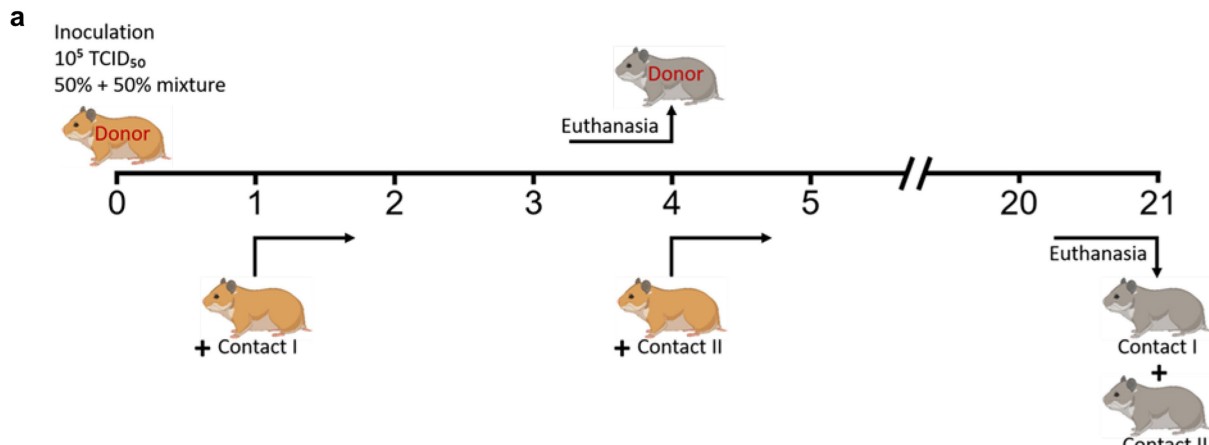

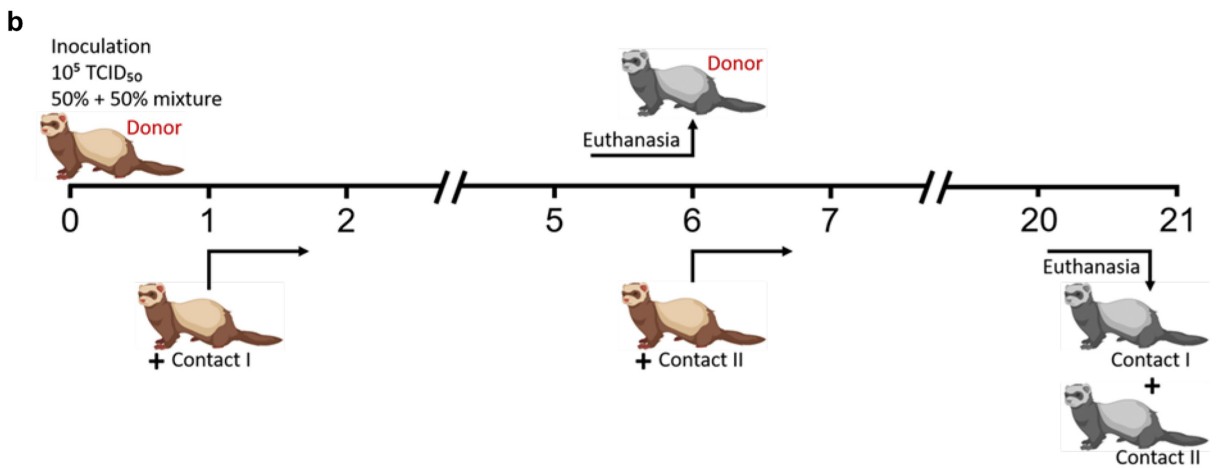

**Extended Data Fig. 2 | Experimental workflow of competitive transmission experiments in Syrian hamsters and ferrets.** (**a**) Timeline of the hamster experiments. Six intranasally inoculated donor hamsters each were co-housed with one naïve contact hamster (1 dpi), building six respective donor-contact I pairs. At 4 dpi, the donor hamsters were euthanized and the initial contact hamsters I were co-housed with one additional hamster (Contact II). (**b**) Timeline of the ferret experiment. The scheme was generated with BioRender (https://biorender.com/).

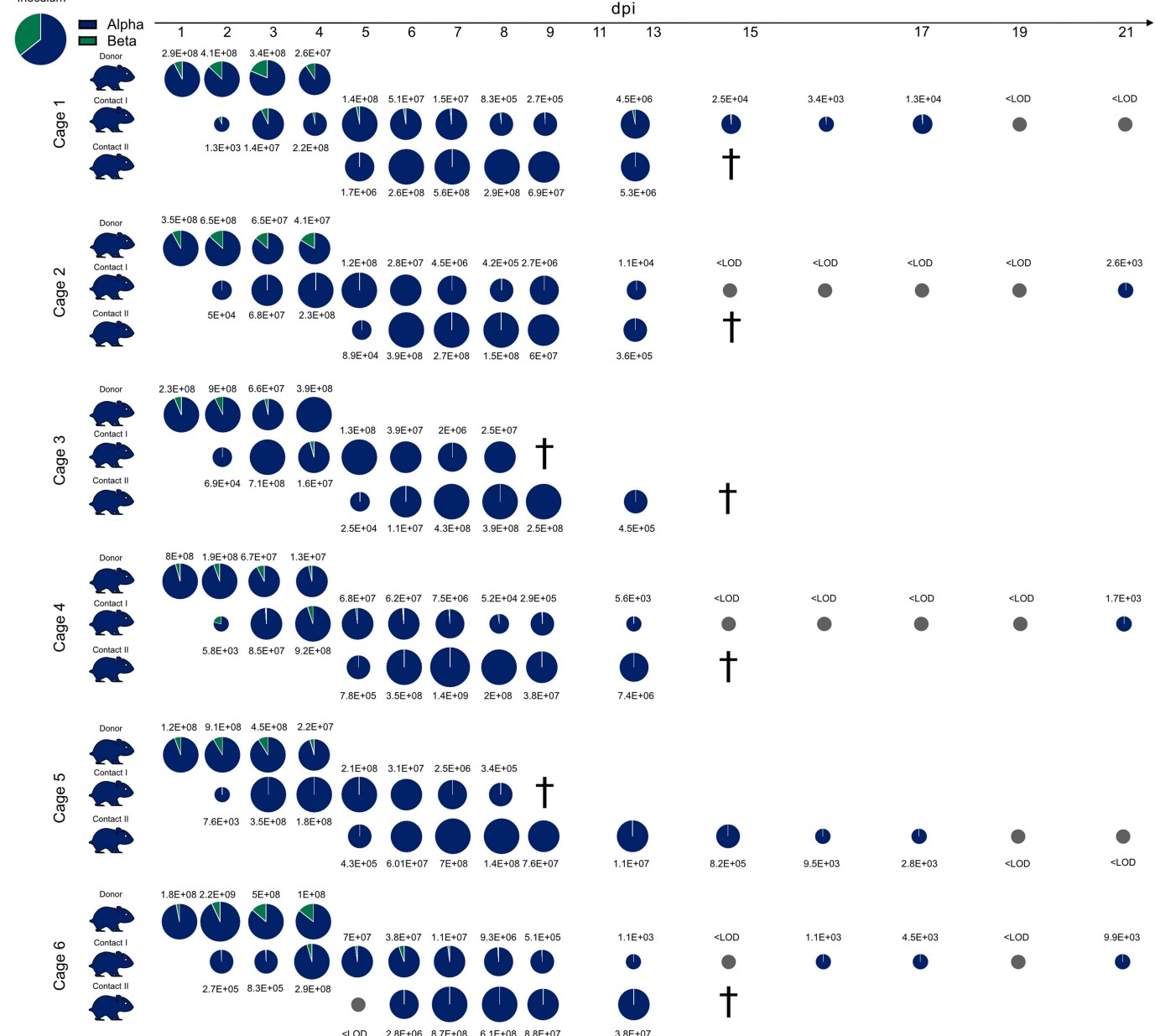

**Extended Data Fig. 3 | Competitive transmission between Alpha and Beta in Syrian hamsters.** Six donor hamsters were each inoculated with $10^{5.06}$ TCID$_{50}$ determined by back titration and composed of a mixture of Alpha (dark blue) and Beta (green) at 1.8:1 ratio determined by RT-qPCR. Donor hamsters, contact I and II hamsters were co-housed sequentially as shown in Extended Data Fig. 2a. Nasal washings were performed daily from 1–9 dpi and afterwards every two days until 21 dpi. Pie chart colors illustrate the ratio of variants detected in nasal washings at the indicated dpi. Pie chart sizes are proportional to the total viral genome copies reported above or below respective pies. Grey pies indicate values below the LOD (<103 viral genome copies per sample). Hamster silhouettes are colored according to the dominant variant (>66%) detected in the latest sample of each animal. † indicate that the corresponding animal reached the humane endpoint.

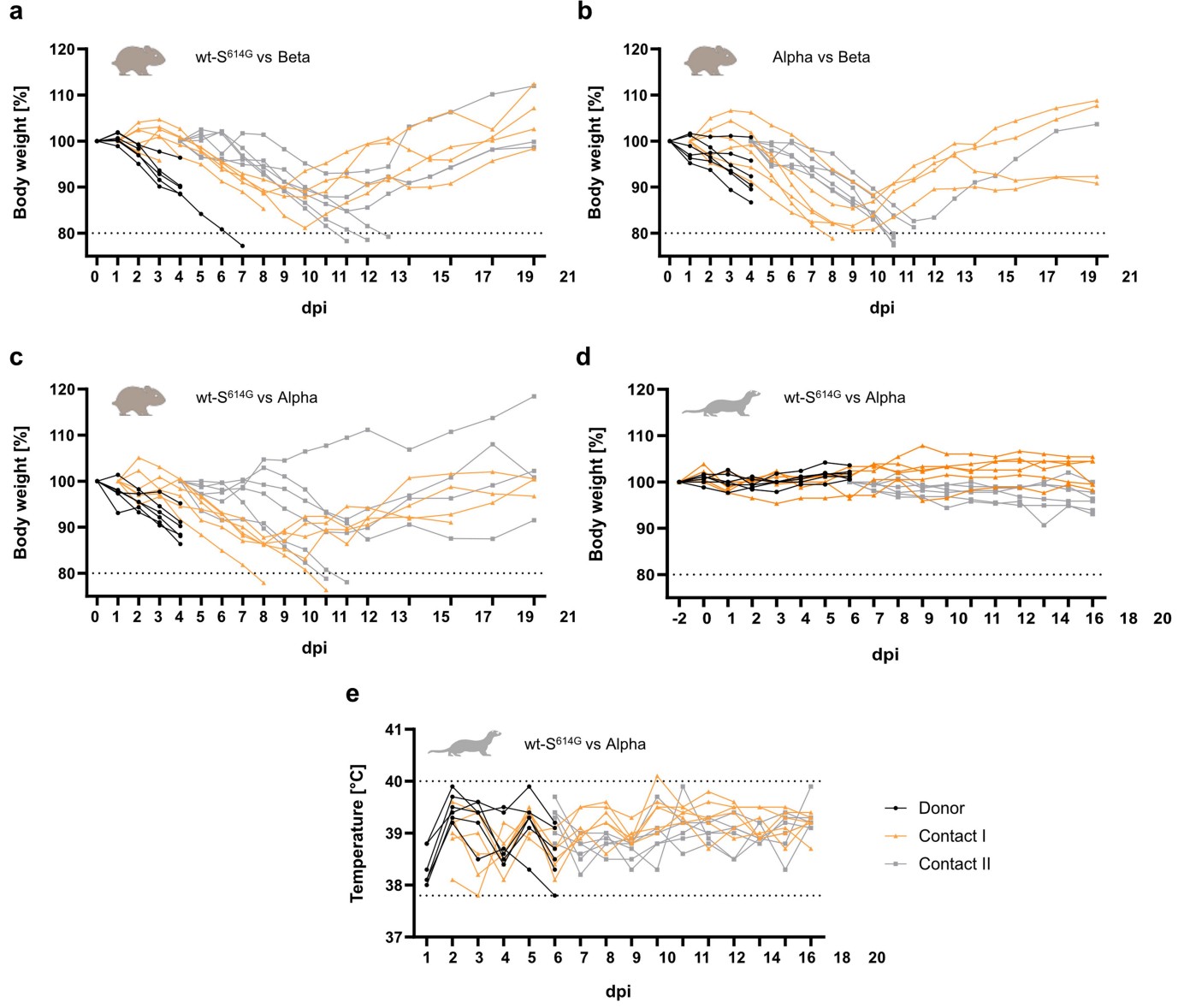

**Extended Data Fig. 4 | Clinical features of hamsters and ferrets. (a–c)** Syrian hamsters were inoculated with comparable genome equivalent mixture of either wt-S$^{614G}$ and Beta (**a**), Alpha and Beta (**b**), or wt-S$^{614G}$ and Alpha (**c**). In hamsters, body weight was monitored daily until 13 dpi, afterwards every two days until 21 dpi and plotted relative to bodyweight of day 0. The dotted line indicates the humane endpoint criterion of 20% body weight loss from initial bodyweight at which hamsters were promptly euthanized for animal welfare reasons. (**d**, **e**) Ferrets were inoculated intranasally with an equal mixture of wt-S$^{614G}$ and Alpha. Body weight (**d**) and temperature (**e**) were monitored daily in ferrets until 12 dpi, and afterwards every 2 days. Grey dotted lines in e indicate the physiologic range for body temperature in ferrets.

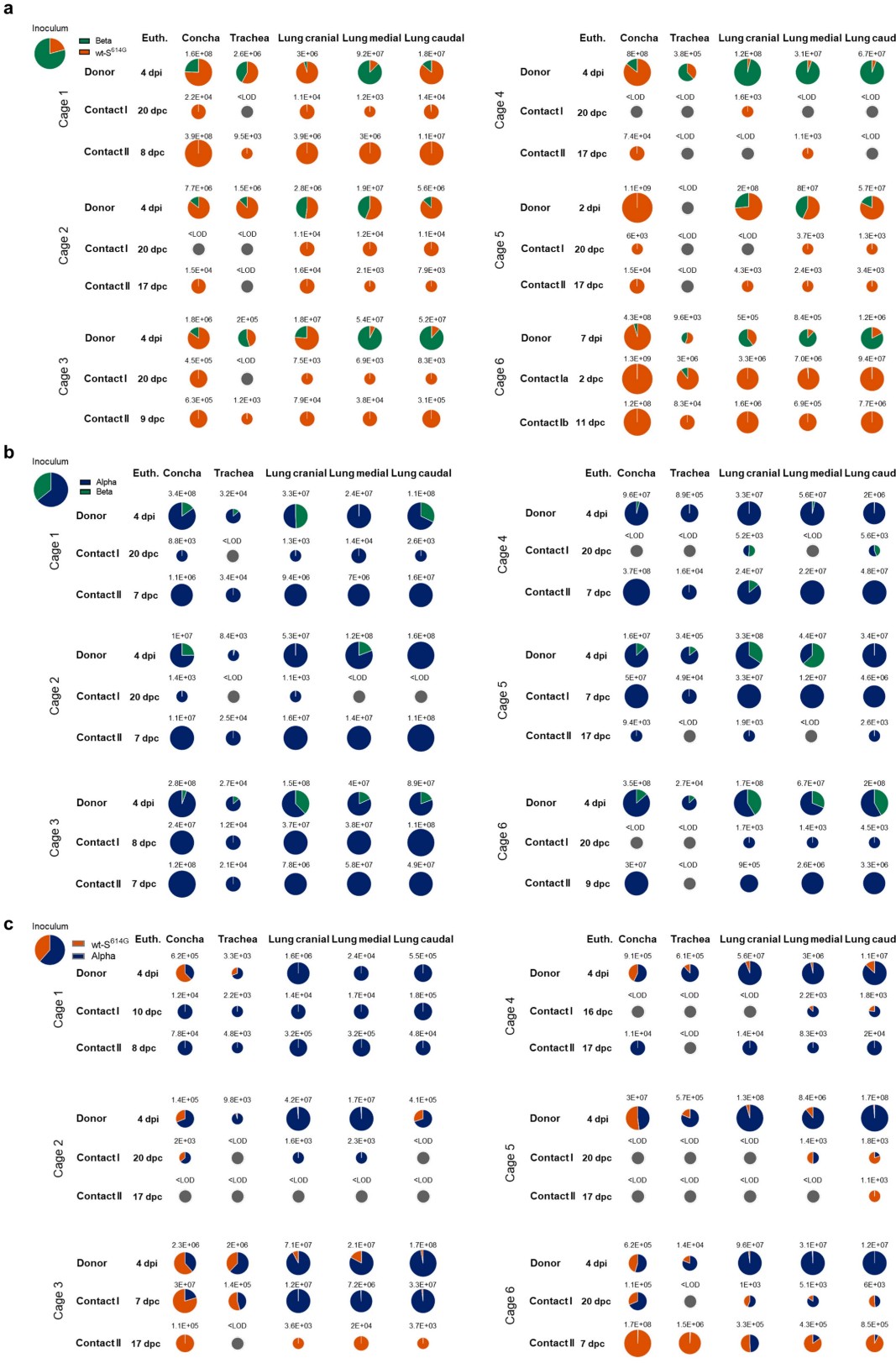

**Extended Data Fig. 5 | Viral genome load in upper (URT) and lower (LRT) respiratory tract tissues of Syrian hamsters in the competitive transmission experiment between SARS-CoV-2 VOCs. (a–c)** Syrian hamsters were inoculated with comparable genome equivalent mixture of either wt-S$^{614G}$ and Beta (**a**), Alpha and Beta (**b**), or wt-S$^{614G}$ and Alpha (**c**). Absolute quantification was performed by RT–qPCR analysis of tissue homogenates of donor, contact I and contact II hamsters in relation to a set of defined standards. Tissue samples were collected at euthanasia (Euth.). Pie chart colors illustrate the ratio of variants detected in each sample at the indicated dpi or days post contact (dpc). Pie chart sizes are proportional to the total viral genome copies reported below. Grey pies indicate values below the LOD (<10$^3$ viral genome copies per sample).

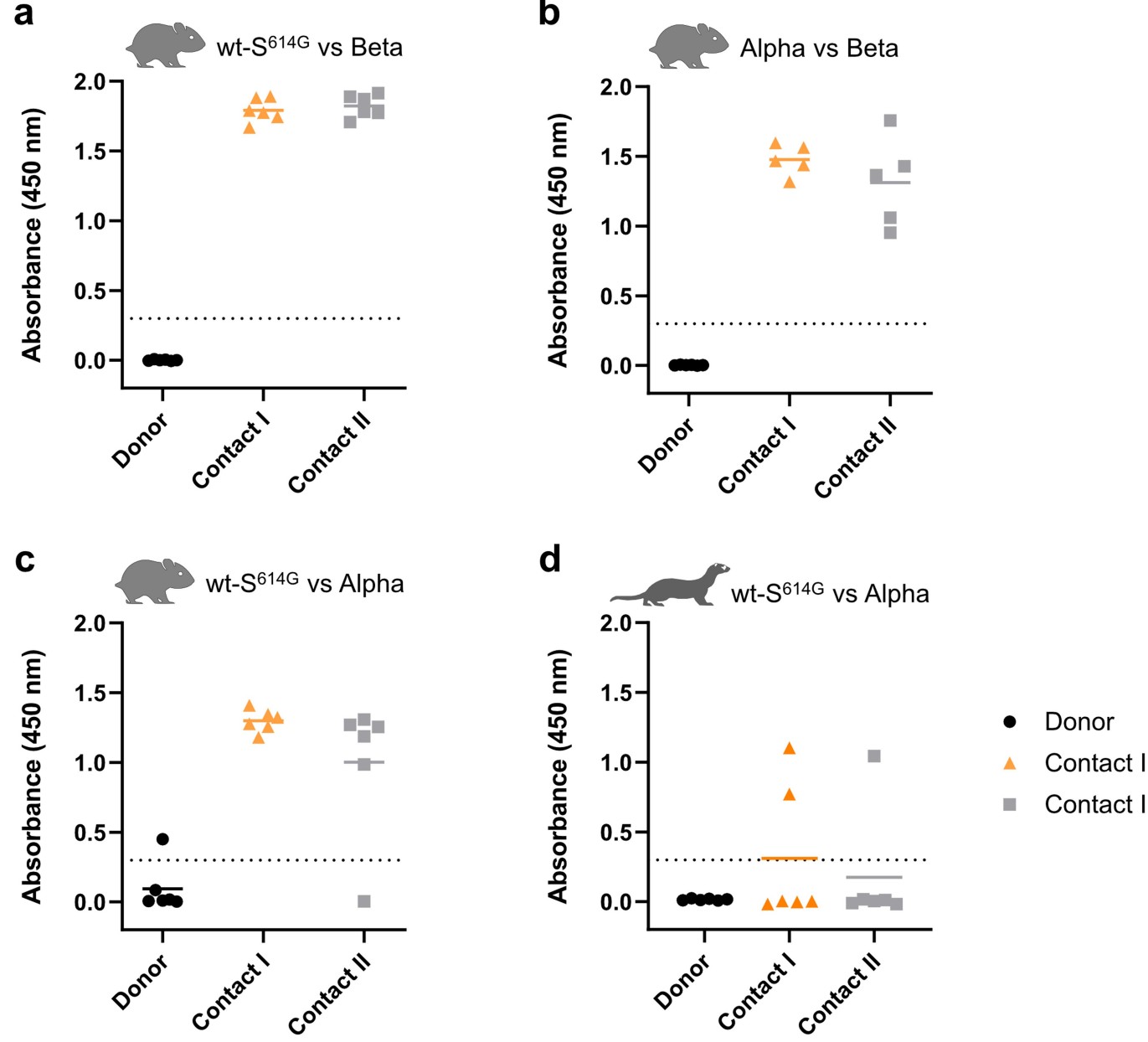

**Extended Data Fig. 6 | Indirect ELISA against the RBD of SARS-CoV-2.** Sera of donor hamsters (**a**, **b**, **c**) and ferrets (**d**) inoculated with the indicated SARS-CoV-2 VOC mixtures and sera of contact I and II animals were collected at their respective experimental endpoints. All sera were tested for specific reactivity against the SARS-CoV-2 RBD-SD1 domain (wt-S amino acids 319-519).

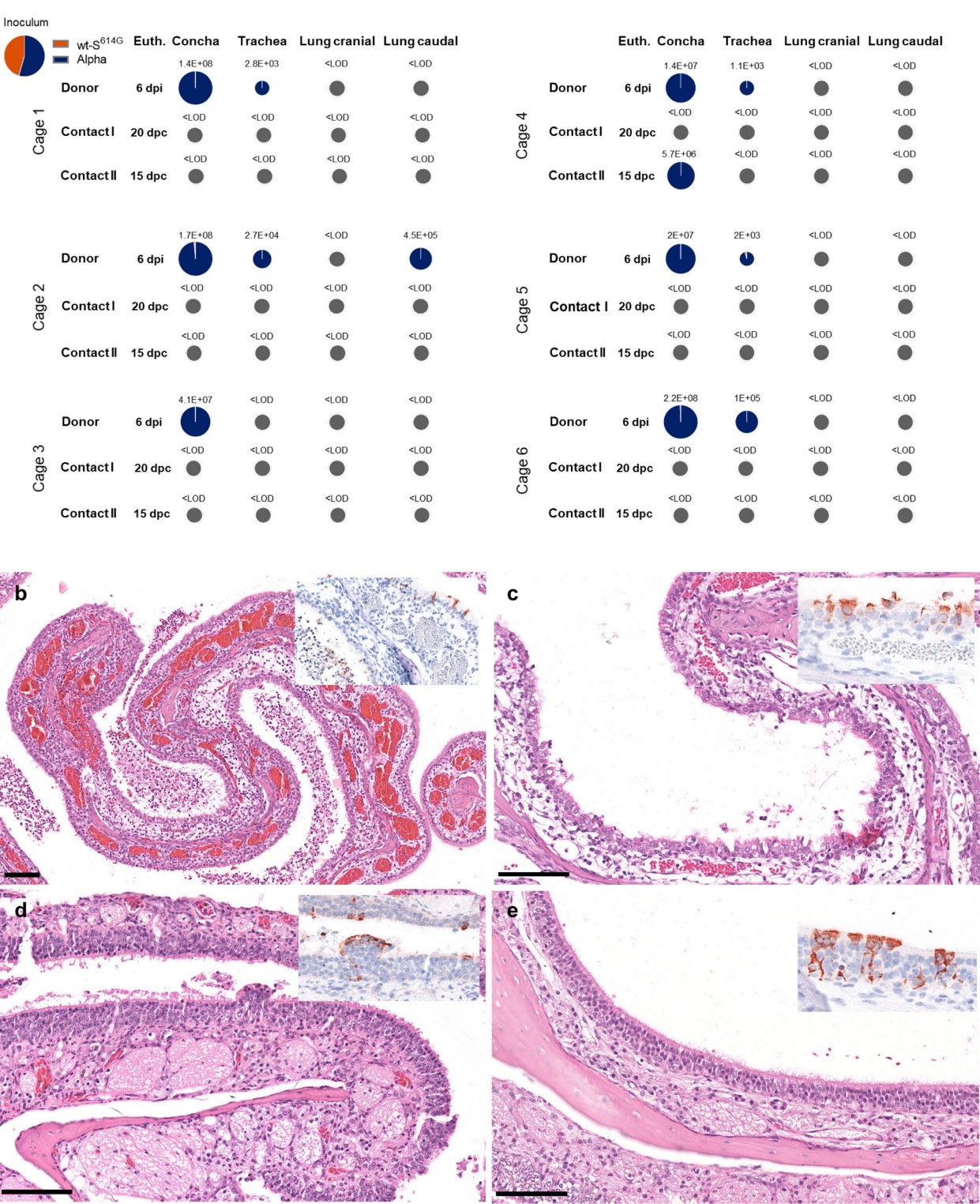

**Extended Data Fig. 7 | Viral genome load in upper (URT) and lower (LRT) respiratory tract tissue of ferrets in the competitive transmission experiment between SARS-CoV-2 Alpha and wt-S^614G.** (**a**) Absolute quantification was performed by RT–qPCR analysis of tissue homogenates of donor, contact I and contact II ferrets in relation to a set of defined standards. Tissue samples were collected at euthanasia (Euth.). Pie chart colors illustrate the ratio of variants detected in each sample at the indicated dpi or dpc. Pie chart sizes are proportional to the total viral genome copies reported below. Grey pies indicate values below the LOD (<10³ viral genome copies per sample).

(**b**–**e**) Representative micrographs of hematoxylin and eosin staining of 3 µm sections of nasal conchae of donor ferrets (n = 6) 6 dpi with wt-S^614G and Alpha. Micrographs are representative of 5 consecutive tissue samples of each animal. Insets show immunohistochemistry staining of SARS-CoV-2 with anti-SARS nucleocapsid antibody with hematoxylin counterstain. The respiratory (**b**, **c**) and olfactory (**d**, **e**) nasal mucosa exhibited rhinitis with varying severity. Lesion-associated antigen was found in ciliated cells of the respiratory epithelium (**b**, **c**) and in sustentacular cells of the olfactory epithelium (**d**, **e**) in all donor animals (n = 6) at 6 dpi. Scale bars are 100 µm.

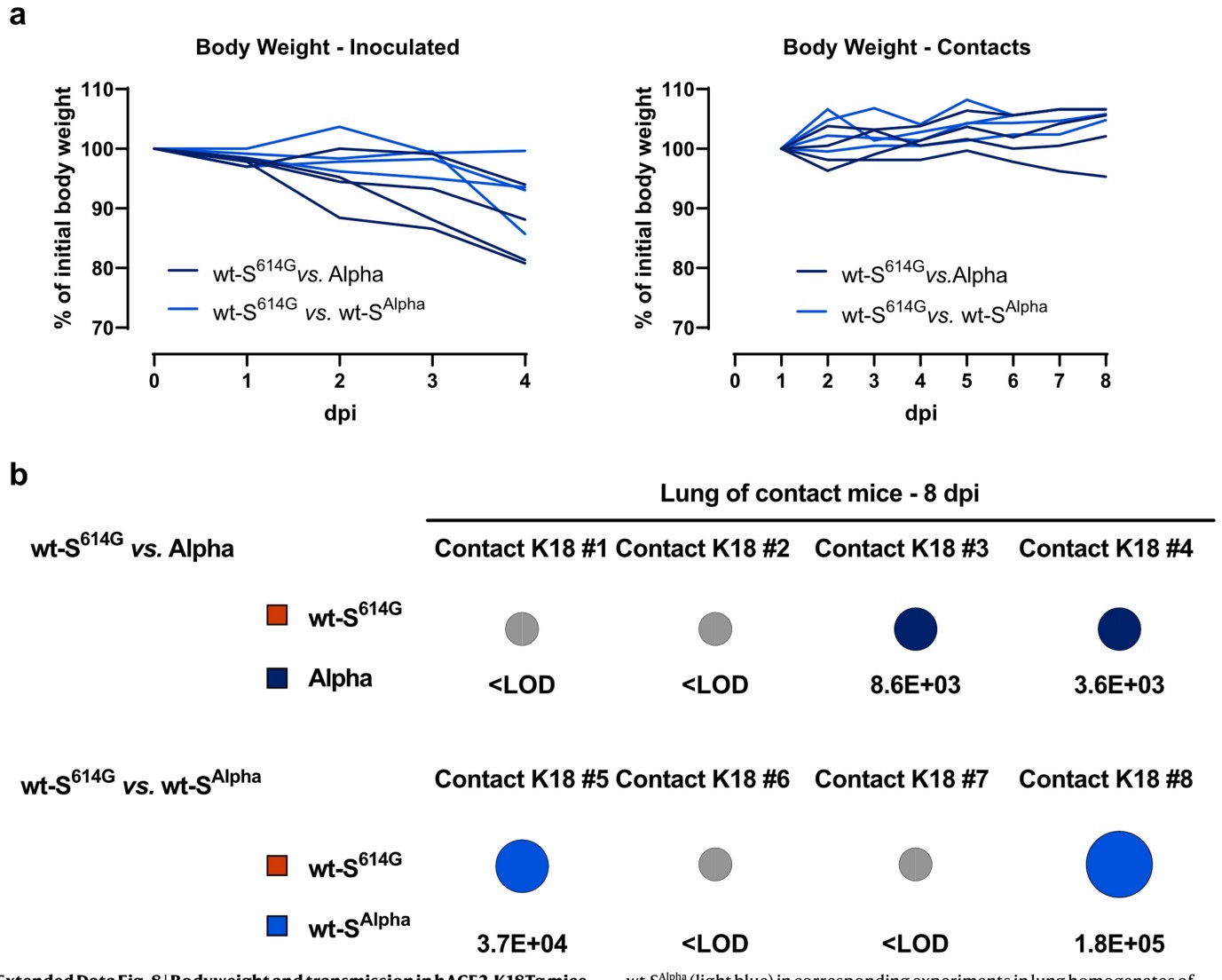

**Extended Data Fig. 8 | Body weight and transmission in hACE2-K18Tg mice.**
hACE2-K18Tg mice inoculated with a mixture of wt-S$^{614G}$ and Alpha, or wt-S$^{614G}$
and wt-S$^{Alpha}$. (**a**) Relative body weight of individual donor mice (n = 4 mice/group;
left panel), and contact mice (n = 4 mice/group; right panel). (**b**) Pie chart
colors illustrate the ratio of wt-S$^{614G}$ (orange) with Alpha (dark blue), or with
wt-S$^{Alpha}$ (light blue) in corresponding experiments in lung homogenates of
contact mice at 7 dpc (i.e., 8 dpi of donor mice). Pie chart sizes are proportional
to the total viral genome copies reported below. Grey pies indicate values
below the LOD (<10$^3$ viral genome copies per sample).

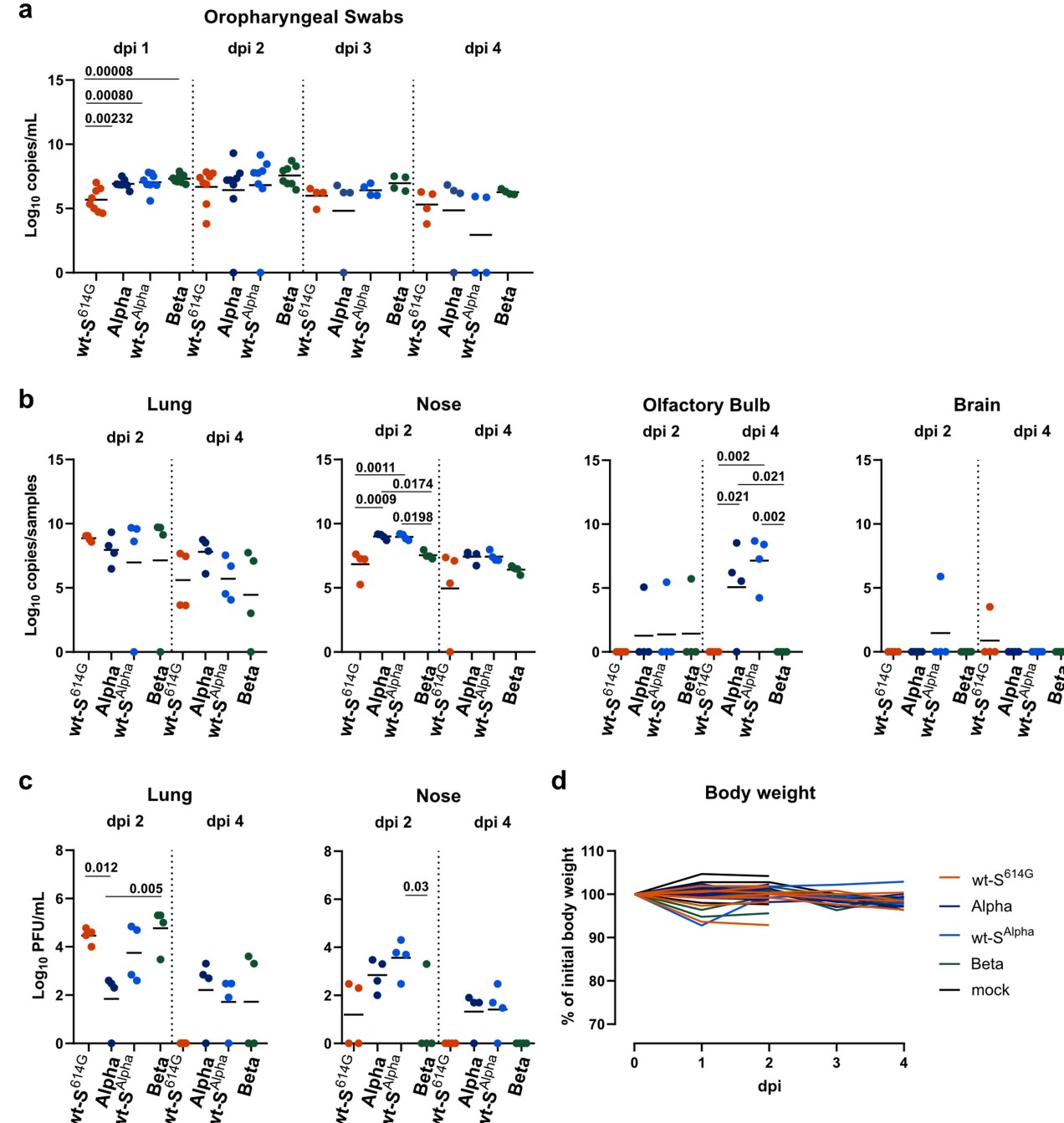

**Extended Data Fig. 9 | Replication of VOC in hACE2-KI mice.** (**a**–**d**) Groups hACE2-KI male mice were inoculated intranasally with $10^4$ PFUs of SARS-CoV-2 wt-$S^{614G}$, Alpha, wt-$S^{Alpha}$ and Beta (n = 8 mice/group). Genome copy numbers in daily oropharyngeal swabs (**a**) and in tissues (**b**), and virus titers (**c**) in tissues were determined at indicated dpi. Data were log10 transformed and presented as individual values and mean. * p<0.05, **p<0.01 by one-way ANOVA with Tukey's multiple comparisons test comparing the four groups. (**d**) Relative body weight of individual hACE2-KI mice overtime relative to weight at infection (n = 8 mice/group until 2 dpi, and n = 4 mice/group from 3 dpi).

## SARS-CoV-2 wt-S^614G

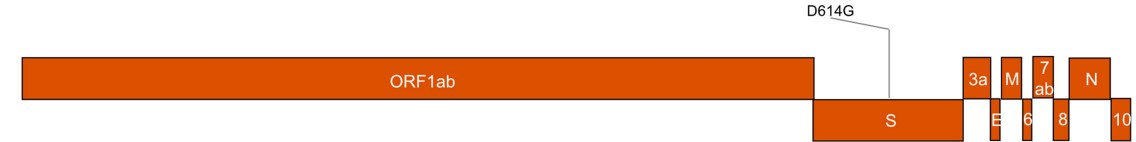

## SARS-CoV-2 Beta

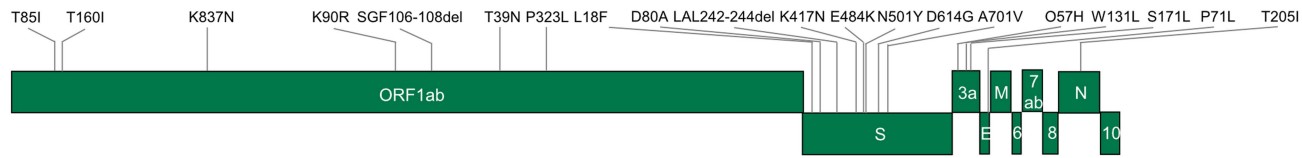

## SARS-CoV-2 Alpha

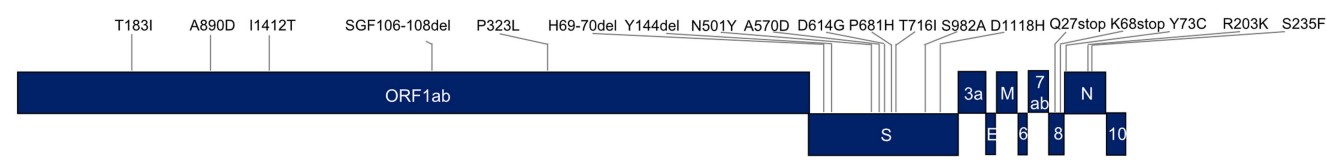

## SARS-CoV-2 wt-S^Alpha

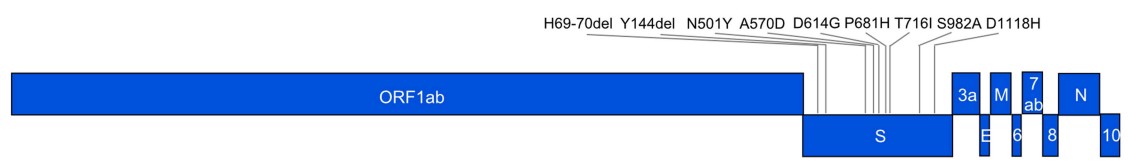

**Extended Data Fig. 10 | Genome sequences of used SARS-CoV-2 variants.** Colors of the variants represent respective viruses in the different experiments. Grey lines indicate positions of known mutations of each virus strain.

**Extended Data Table 1 | Sequence mutations in S in SARS-CoV-2 recombinant strains**

| SARS-CoV-2 viruses | Mutations in S | Use | Accession ID (BioProject / GenBank / GISAID) |
|---|---|---|---|
| **Isolates:** | | | |
| wt-S$^{614G}$ (B.1.610) | D614G | Kinetics in vitro | OL675863, EPI_ISL_414019 |
| Alpha | H69/V70del, Y144del, N501Y, A570D, D614G, P681H, T716I, S982A, D1118H | Kinetics in vitro | OL689430, EPI_ISL_2131446 |
| Beta | L18F, D80A, D215G, L242/A243/L244del, K417N, E484K, N501Y, D614G, A701V, | Kinetics in vitro | OL689583, EPI_ISL_981782 |
| Alpha (L4549) | H69/V70del, Y144del, N501Y, A570D, D614G, P681H, T716I, S982A, D1118H | Competition wt-S$^{614G}$ vs Alpha; in vitro, hamster, ferret, and mice<br>Individual virus infection, mice | PRJEB45736, SARS-CoV-2 B.1.1.7 NW-RKI-I-0026/2020 passage 3 L4549, EPI_ISL_803957 |
| Beta (L4550) (Ref. 21) | L18F, D80A, D215G, L242/L243/A244del K417N, E484K, N501Y, D614G, A701V | Competition wt-S$^{614G}$ vs Beta; hamster, mice<br>Competition Alpha vs Beta; hamster | MZ433432, L4550, EPI_ISL_751799 |
| **Recombinant clones:** | | | |
| wt-S$^{614G}$ (Ref. 3) | D614G | Competition wt-S$^{614G}$ vs Beta; in vitro, hamster, mice<br>Competition wt-S$^{614G}$ vs Alpha; in vitro, mice<br>Competition wt-S$^{614G}$ vs wt-S$^{Alpha}$; mice<br>Single virus infection, mice | MT108784 |
| wt-S$^{614G}$ (*) (L4595) | D614G, R685S | Competition wt-S$^{614G}$ vs Alpha; hamster and ferret | PRJEB45736, wt-S614G ID#49 vial 2, L4595 |
| wt-S$^{Alpha}$ | H69/V70del, Y144del, N501Y, A570D, D614G, P681H, T716I, S982A, D1118H | Competition wt-S$^{614G}$ vs wt-S$^{Alpha}$; mice<br>Single virus infection, mice | PRJNA784099 |

(*) SARS-CoV-2 wt-S$^{614G}$ used for the competitive experiments between wt-S$^{614G}$ vs. Alpha in hamsters and ferrets had an exchange in 54% of the analyzed contigs from C to A in position 23615, located in the S gene.

**Extended Data Table 2 | Sequences of primer and probes for RT-qPCR assays**

| Oligo name | Sequence (5´- 3´) | Conc. | Position |
|---|---|---|---|
| **SARS-CoV-2 wt-S$^{614G}$ -ORF1 assay** | | | |
| **wt-S$^{614G}$ -ORF1-1.2F** | TGG TTG ATA CTA GTT TGT CTG GT | 10 μM | 11271-11293 * |
| **wt-S$^{614G}$ -ORF1-1.3R** | GCA CCA TCA TCA TAC ACA GTT C | 10 μM | 11379-11358 * |
| **wt-S$^{614G}$ -ORF1-1FAM** | FAM-TGC ATC AGC TGT AGT GTT ACT AAT CC-BHQ1 | 5 μM | 11320-11345 * |
| **SARS-CoV-2 wt-S$^{614G}$ -S assay** | | | |
| wt-S$^{614G}$ -S-1.2F | TAC TTG GTT CCA TGC TAT ACA TGT | 10 μM | 21748-21771 * |
| wt-S$^{614G}$ -S-1.5R | CCA ACT TTT GTT GTT TTT GTG GTA ATA | 10 μM | 22018-21992 * |
| wt-S$^{614G}$ -S-1FAM | FAM-ACC CTG TCC TAC CAT TTA ATG ATG-BHQ1 | 5 μM | 21804-21827 * |
| **SARS-CoV-2 Alpha -ORF1 assay** | | | |
| B117-ORF1-2.1F | GAT ATG GTT GAT ACT AGT TTG AAG | 10 μM | 11265-11288 ** |
| **wt-S$^{614G}$ -ORF1-1.3R** | See above | | |
| **wt-S$^{614G}$ -ORF1-1FAM** | See above | | |
| **SARS-CoV-2 Alpha -S assay** | | | |
| B117-S-1.2F | TTA CTT GGT TCC ATG CTA TMT C | 10 μM | 21736-21757 ** |
| B117-S-1.3R | AAC TTT TGT TGT TTT TGT GGT AAA C | 10 μM | 21996-21972 ** |
| wt-S$^{614G}$ -S-1FAM | See above | | |

* Position based on NC_045512; ** Position based on MW963651. Conc, concentration.

**Extended Data Table 3 | Attribution of RT-qPCR assays used for the individual competitive transmission experiments**

| Assay | For detection of |
|---|---|
| **wt-S$^{614G}$ vs Alpha (hamsters, ferrets)** | |
| SARS-CoV-2 wt-S$^{614G}$ -ORF1 assay | SARS-CoV-2 wt-S$^{614G}$ |
| SARS-CoV-2 Alpha -ORF1 assay | SARS-CoV-2 Alpha |
| **wt-S$^{614G}$ vs Alpha (human AEC cultures, mice)** | |
| SARS-CoV-2 wt-S$^{614G}$ -S assay | SARS-CoV-2 wt-S$^{614G}$ |
| SARS-CoV-2 Alpha -S assay | SARS-CoV-2 Alpha |
| SARS-CoV-2 Alpha -S assay | SARS-CoV-2 wt-S$^{Alpha}$ |
| **wt-S$^{614G}$ vs Beta (human AEC cultures, hamsters and mice)** | |
| SARS-CoV-2 wt-S$^{614G}$ -ORF1 assay | SARS-CoV-2 wt-S$^{614G}$ |
| SARS-CoV-2 Alpha -ORF1 assay | SARS-CoV-2 Beta |
| **Alpha vs Beta (human AEC cultures, hamsters)** | |
| SARS-CoV-2 Alpha -S assay | SARS-CoV-2 Alpha |
| SARS-CoV-2 wt-S$^{614G}$ -S assay | SARS-CoV-2 Beta |

Martin Beer
Volker Thiel

# Reporting Summary

## Statistics

For all statistical analyses, confirm that the following items are present in the figure legend, table legend, main text, or Methods section.

| n/a | Confirmed | |
|---|---|---|
| ☐ | ☒ | The exact sample size (*n*) for each experimental group/condition, given as a discrete number and unit of measurement |
| ☐ | ☒ | A statement on whether measurements were taken from distinct samples or whether the same sample was measured repeatedly |
| ☐ | ☒ | The statistical test(s) used AND whether they are one- or two-sided *Only common tests should be described solely by name; describe more complex techniques in the Methods section.* |
| ☐ | ☒ | A description of all covariates tested |
| ☐ | ☒ | A description of any assumptions or corrections, such as tests of normality and adjustment for multiple comparisons |
| ☐ | ☒ | A full description of the statistical parameters including central tendency (e.g. means) or other basic estimates (e.g. regression coefficient) AND variation (e.g. standard deviation) or associated estimates of uncertainty (e.g. confidence intervals) |
| ☐ | ☒ | For null hypothesis testing, the test statistic (e.g. $F$, $t$, $r$) with confidence intervals, effect sizes, degrees of freedom and $P$ value noted *Give P values as exact values whenever suitable.* |
| ☒ | ☐ | For Bayesian analysis, information on the choice of priors and Markov chain Monte Carlo settings |
| ☒ | ☐ | For hierarchical and complex designs, identification of the appropriate level for tests and full reporting of outcomes |
| ☒ | ☐ | Estimates of effect sizes (e.g. Cohen's *d*, Pearson's *r*), indicating how they were calculated |

*Our web collection on statistics for biologists contains articles on many of the points above.*

## Software and code

Policy information about availability of computer code

| Data collection | ELISA: Tecan i-control 2014 1.11 qRT-PCR: QuantStudio™ Real-Time PCR Software (v1.7.1), or 7500 Fast System SDS Software Version 1.4 Viral titers: manual counting, registered in Microsoft Excel  2016 (16.0.5239.1001) BLI: Octet RED96e instrument with ForteBio Data Acquisition Software (Version: 12.0.1.8) |
|---|---|
| Data analysis | relative variant quantification: Bio-Rad CFX Maestro 1.1 Version 4.1.2433.1219 sequence analysis: Geneious Prime ® 2019.2.3 figures: GraphPad Prism 8.4.2 (679) for Windows, Microsoft PowerPoint 2016 (16.0.4266.1001), Adobe Photoshop CS5 64 bit NGS: Genome Sequencer Software Suite (version 2.6; Roche, https://roche.com), variant analysis tool integrated in Geneious Prime (2019.2.3) digital PCR: QuantaSoft Analysis Pro software (version 1.0.596) ELISA: Microsoft Excel 2016 (16.0.5188.1000) Statistical analysis: GraphPad Prism version 8 or R (version 4.1), using the packages tidyverse (v1.3.1), ggpubr (v0.4.0), rstatix (v.0.7.0). BLI: ForteBio Data Analysis Software (Version 12.0.1.2) |

For manuscripts utilizing custom algorithms or software that are central to the research but not yet described in published literature, software must be made available to editors and reviewers. We strongly encourage code deposition in a community repository (e.g. GitHub). See the Nature Portfolio guidelines for submitting code & software for further information.

## Data

Policy information about availability of data

All manuscripts must include a data availability statement. This statement should provide the following information, where applicable:

- Accession codes, unique identifiers, or web links for publicly available datasets
- A description of any restrictions on data availability
- For clinical datasets or third party data, please ensure that the statement adheres to our policy

Sequence data are available on the NCBI Sequence Read Archive (SRA) under the accession numbers PRJEB45736, and PRJNA784099, or in GenBank under the accession numbers MT108784, MZ433432, OL675863, OL689430, and OL689583 as shown in Extended Data Table 1. Source data are provided with this paper.

# Field-specific reporting

Please select the one below that is the best fit for your research. If you are not sure, read the appropriate sections before making your selection.

☒ Life sciences  ☐ Behavioural & social sciences  ☐ Ecological, evolutionary & environmental sciences

For a reference copy of the document with all sections, see nature.com/documents/nr-reporting-summary-flat.pdf

# Life sciences study design

All studies must disclose on these points even when the disclosure is negative.

| Sample size | No sample size calculations were performed. Number of animals used in experiments were based on our previous comparative studies of fitness of SARS-CoV-2 variants. |
|---|---|
| Data exclusions | No data were excluded from analysis. |
| Replication | Binding Assays (BLI): data are representative of 3 independent experiments.<br>Airway epithelial cell (AEC) cultures in vitro: All attempts at replication were successful; experiments were performed independently on different biological donors according to best practices and as described in the Methods.<br>In vivo: Competition experiments between two VOCs were performed on groups of 6 infected animals and replicated in 4 animal models. Competition between wt-S614G and Alpha was replicated in males and females. All attempts at replication were successful. Single VOC infections were performed in groups of 8 mice for each variant. |
| Randomization | No randomization was required for all in vitro and in vivo competition experiments because the viral and host response parameters were measured within each cell culture insert or each animal.<br>For single infection with VOCs, hACE2-KI were randomly assigned to the respective study groups. |
| Blinding | Investigators were blinded during analysis of viral plaque and qRT-PCR assays of in vitro and in vivo experiments.<br>Blinding was also not relevant for in vivo competition experiments as both arms of the comparison were in single animals. |

# Reporting for specific materials, systems and methods

We require information from authors about some types of materials, experimental systems and methods used in many studies. Here, indicate whether each material, system or method listed is relevant to your study. If you are not sure if a list item applies to your research, read the appropriate section before selecting a response.

## Materials & experimental systems

| n/a | Involved in the study |
|---|---|
| ☐ | ☒ Antibodies |
| ☐ | ☒ Eukaryotic cell lines |
| ☒ | ☐ Palaeontology and archaeology |
| ☐ | ☒ Animals and other organisms |
| ☒ | ☐ Human research participants |
| ☒ | ☐ Clinical data |
| ☒ | ☐ Dual use research of concern |

## Methods

| n/a | Involved in the study |
|---|---|
| ☒ | ☐ ChIP-seq |
| ☒ | ☐ Flow cytometry |
| ☒ | ☐ MRI-based neuroimaging |

## Antibodies

| Antibodies used | anti-SARS nucleocapsid antibody (Novus Biologicals #NB100-56576) |
|---|---|

| Validation | relevant validation information can be accessed at https://www.novusbio.com/products/sars-nucleocapsid-protein-antibody_nb100-56576#reviews-publications |
|---|---|

# Eukaryotic cell lines

Policy information about cell lines

| Cell line source(s) | Vero E6 cells (FLI): Collection of Cell Lines in Veterinary Medicine CCLV RIE 0929<br>Vero E6 cells (IVI, IFIK): cells were kindly provided by Doreen Muth, Marcel Müller, and Christian Drosten, Charité, Berlin, Germany (ATCC CRL-1586)<br>Vero-TMPRSS2 cells were kindly provided by Stefan Pöhlmann, German Primate Center - Leibniz Institute for Primate Research, Göttingen, Germany)<br>Primary human nasal cells were commercially procured from Epithelix, in Geneva, Switzerland.<br>Expi293F cells: ThermoFisher Scientific, USA.<br>A549-hACE2 cells were derived from ATCC CCL-185 and kindly provided by M. Schmolke, B. Mazel-Sanchez, and F. Abdul, Faculty of Medicine, Geneva |
|---|---|
| Authentication | in-house authentication for cell lines was not performed |
| Mycoplasma contamination | Cell lines were tested negative for mycoplasma contamination. |
| Commonly misidentified lines<br>(See ICLAC register) | None |

# Animals and other organisms

Policy information about studies involving animals; ARRIVE guidelines recommended for reporting animal research

| Laboratory animals | Mustela putorius furo, ferrets, neutered male and female, 8 - 23 months<br>Mesocricetus auratus, Syrian hamster, male, 7-12 weeks<br>Mus musculus, mice B6.Cg-Tg(K18-ACE2)2Prlmn/J, male and female, 10-12 weeks<br>Mus musculus, mice B6.Cg-Ace2<tm1(ACE2)Dwnt>, male and female, 10-12 weeks |
|---|---|
| Wild animals | No wild animals were used |
| Field-collected samples | Field samples were not collected |
| Ethics oversight | Ferrets/hamsters: State Office of Agriculture ethics committee, Food Safety, and Fishery in Mecklenburg–Western Pomerania, Germany, registration number LVL MV TSD/7221.3-1-004/21<br>Mice: Commission for Animal Experimentation, Cantonal Veterinary Office of Bern, Switzerland, license BE-43/20 |

Note that full information on the approval of the study protocol must also be provided in the manuscript.

