## [Peer Review File · Nature]

Manuscript Title: Enhanced fitness of SARS-CoV-2 variant of concern Alpha, but not Beta

Reviewer Comments & Author Rebuttals

Reviewer Reports on the Initial Version:

Referee #1 (Remarks to the Author):

This is a well constructed and well written manuscript demonstrating that the alpha variant transmits more readily between hamsters than does the beta variant. The report is important because it provides an explanation for why the beta variant, which is the variant with the greatest ability to evade the anti-SARS-CoV-2 immune response, has not outgrown the alpha variant and become dominant. The manuscript also compares transmission of the alpha and S614B strains in mice and ferrets.

General comments.

1. While the manuscript describes interesting data, it is somewhat out of date, as the delta variant has become more dominant and has fueled the recent surge in new cases. If any comparison data between the alpha and delta variants are available, they should be included. It is recognized that the emergence of new variants is faster at this phase of the pandemic than is the ability to analyze their relative transmissibility, but such a comparison would enhance the manuscript.
2. Page 7, para 2-It is curious that there is no fitness difference between the alpha variant and wt-S614G virus in hamsters but not in the other species that were analyzed. Differences might be more apparent if indirect transmission studies were added to the contact transmission studies.

Additional comments.

1. Both qRT-PCR and nextgen sequence analyses are described in the methods. Which one was used to generate the comparative sequence data shown in the figures?
2. ED Figures 7,8, page 7, para 1-The beta variant becomes dominant in the lungs of some dually infected donor hamsters. At the same inoculum, is the beta variant more lethal than the alpha or S614G virus (this may be known from the literature already)?
3. Page 9, para 2, Figure 5. The presence of encephalitis in the hACE2-K18Tg mice may reflect the use of young mice and a high dose of virus. This does not change the results but should be noted.
4. Page 9, 2 lines from bottom-Should be Fig. 5B.
5. Page 10-first 4 lines-While the authors are properly cautious in interpreting the results, it seems equally likely that the results reflect small numbers of animals that were analyzed.
6. Page 10, 2 lines from bottom-should be Fig. 6E,F.

Referee #2 (Remarks to the Author):

The authors have assembled a truly impressive amount of data in in vitro and in vivo models to assess the fitness of SARS-CoV-2 variants including the alpha and beta variants compared with a prototypic earlier wave virus.

They employ both natural virus isolates and recombinant viruses engineered by reverse genetics to contain one or more Spike gene mutations present from the alpha VOC.

Having one or more experimental models by which new and emerging variants could be risk assessed would be incredibly valuable for making public health decisions. Applying a reverse genetics approach to dissect which mutations in VOCs (that generally carry constellations of mutations) would also reveal valuable mechanistic information by which future variants can be assessed using sequence information alone, and by which we can better understand how this virus infects, causes disease and transmits.

Overall the work is thus highly topical, of huge interest and importance.

However, the manuscript as presented currently lacks coherence. There is a lot of data, but none

of it adds up to one single story and each part is somewhat incomplete.

For example, the authors show that the human airway cultures do not reveal any differences between the replication of the alpha variant, or a recombinant virus bearing the alpha spike and the wild type precursor. Yet they go one to test replication of single point spike mutations in the airway cultures- it is not clear why they hypothesize they might detect differences in those mutants when the whole alpha variant isolate or spike recombinant did not show this. And indeed they find no differences between the mutants.

On the other hand, they do show that in the ferret model and in the human ACE2 mouse there is a fitness advantage of the alpha variant- but in these models they do not test their set of single point recombinants. In other words the recombinants have not really been put to use.

Similarly they do show in the hamster model an advantage of alpha over beta but they do not test this in any other model, even not the in vitro cultures.

They include a lot of data in supplementary figures that are not really essential to their story and duplicate work that has been performed and published extensively elsewhere by others such as neutralization data around alpha and beta VOCs.

On the other hand, they do perform some affinity assays using the ACE2 proteins from the various species they test, with wild type and alpha spike (but not beta) which could be really key in understanding why some animal models are more appropriate for measuring difference in VOCs than others, but this is a supp figure and hardly discussed, and the beta spike is not tested against hamster ACE2, the animal in which they did find a fitness difference.

Finally the whole presentation is rather haphazard with inconsistent use of log or non log viral titres, two different methods for neutralization assays, lack of inclusion of inoculum ratios for competition assays in some but not all figures, and an array of various statistical analyses some of which do not seem appropriate (details below).

To pull the data all together the authors conclude in this manuscript it is a good idea to use a lot of different model to assess variants. This is a useful point, but in this paper they do not cross compare with the models they have employed enough to really illustrate why one needs more models and which might be preferred, or why.

Specific comments:

Figure 1. illustrates in vitro work using primary cultures of human airway cells

In A the authors employ nasal epithelia cells and measure viral RNA loads over time.

There are only 2 replicates, and yet they plot mean and sd. Probably more appropriate to simply plot the data points. Are these data repeatable in cultures from different donors? Could more wells have been used?

In B they plot virus infectivity and only analyse the last time point by which time one can see from part A that exponential growth has finished. Differences might be much more pronounced at an earlier time point. These experiments should be repeated and more appropriate time points analysed by both RNA and infectious yield.

The next set of in vitro experiment employs a competition assay in primary airway cells. Figure 1 C simply illustrates the genotype mix in the inoculum. This is not a real result and does not warrant its own figure, suggest incorporate C into part D.

In part D we see the results of the competition experiments, B117 vs wt using two different B117 isolates, and also using a spike chimera

It is not clear which statistical tests have been used here? The figure makes the wild type look fitter than B117?? Eg 120 hours lower left panel

Are these really mean and SD? These are shown as box plots, And they are not symmetric (even taking into account the log scale)?

Extended figure 1 shows virus replication kinetics with recombinant viruses harbouring single point mutations from the B117 spike. The presentation is really difficult to interpret; each virus is plotted alone -one can't really see if there are any difference, nor whether statistical analyses have been used to test whether there are any differences in replication?

Extended figure 2 shows results of neutralization assays against alpha and beta vs wild type

viruses with quite a limited set of sera. As stated above, this type of data has been shown already by multiple groups it is not novel at all... the authors could easily cite these other studies. Moreover the authors here have used two different assays, with quite different output titres, and it is not clear why. The data presentation is difficult, we can't see the individual results. What are the vertical lines in part A? Overall the data presentation in part B is easier to read.

Figures 2-3 and extended data figure 4 describes the results of Hamster competition assays which utilize a chain of direct contact transmission from a donor animal infected with a mix of viruses, Please show the inoculum ratios in the figure as done for the mouse experiments in figure 5. In the extended figures 7,8,9 the viral load in urt vs lrt are analysed; Extended figure 4 is not very clear cut.. there are clearly some contact animals with beta in lungs, the viral load titres written above each circle are quite difficult to unravel, might this data be better presented in a histogram? Is there really any reason to include extended figure 6 survival curves? for competition experiments how do you interpret these?

In Figure 4 the authors turn to a different animal model: they use 6 sets of ferrets in the same style of competition experiment, a direct contact transmission chain, but this time only between wt vs b117

Infection and transmission in ferrets is much less efficient. Indeed the viral loads are very low. It is not clear whether sampling was stopped at day 6 for the contact 1 ferrets? Or whether all animals of 4 chains remained negative? Since one of the 2 actual transmissions was detected very late (day 5 after exposure) perhaps there were more contact 1 animals that became positive after day 6? The authors should make more clear in figure 4 what the grey circles demonstrate and which samples were collected.

Nonetheless, the interesting conclusion is that in this highly stringent model one can detect fitness advantage of alpha VOC over wild type, this correlates with epidemiological experience and suggests the hamster model as used here not suitable for this type of fitness assessment going forwards.

There is no discussion of whether using eg indirect contact or shorter exposure might make the hamster model more stringent and suitable.

In figure 5 the authors then turn to the mouse models: first the K18 human ACE2 mice which also express mouse ACE2 and then a full humanized mouse where human ACE2 replaces murine ACE2 Like the ferret the mouse model is 'inefficient'; and thus reveals the fitness advantage for B117 in line with human epidemiology.

In part B the RG virus with B117 spike also shows an advantage over whole wild type virus, so here is a point at which perhaps the individual recombinants could have been tested?

The presentation of raw genome data in these figure rather than log values is difficult for readers and inconsistent with earlier style. It is not clear exactly which criteria or statistical tests have been used to decide whether mice a 'blue or orange'

Extended figure 14 suggest some limited transmission in the mice : 2/4 transmissions, with always B117 predominant. These numbers are small, and the viral loads very low. How confident are the authors that this could be a feasible animal model going forwards?

Figure 6 use the humanized ACE2 mice and competes B117 vs wild type virus. Part D has high pfu in the lung for wt, but the genome copies in part C are the same as B117. The data done look entirely matched. Again, there are very low E gene copies and some below LOD. Is the mouse model a reliable model even though it does show up this difference?

Finally there are a string of extended data figure around serological response of the infected animals for example supp figure 10 and 11. Here donor animals are tested- but we are told they were culled at day 4 or day 6- It doesn't seem very logical to test their antibody response? What was the hypothesis here?

Ext fig 10 A does illustrate serological responses that correlate with the transmission events in ferrets but please make clear which days were the terminal end points in each case. In part B would we not expect all hamsters even in contact 2 group to seroconvert?

Ext fig 11 uses neutralization assays to analyses sera from animals after mixed infections. I think it is important here to understand what the extent of cross reaction would be after single infections, otherwise such data is difficult to interpret.

Finally the concept of the hamster as a superspreader is a nice image, however in general superspreaders are individuals who for some reason transmit more efficiently eg due to higher bioaerosol exhalation- there is no measure of this here, rather that hamsters are highly susceptible to SARS-CoV-2. I am not sure the analogy quite works

Author Rebuttals to Initial Comments:

Author response to Referee #1:

This is a well constructed and well written manuscript demonstrating that the alpha variant transmits more readily between hamsters than does the beta variant. The report is important because it provides an explanation for why the beta variant, which is the variant with the greatest ability to evade the anti-SARS-CoV-2 immune response, has not outgrown the alpha variant and become dominant. The manuscript also compares transmission of the alpha and S614B strains in mice and ferrets.

General comments. 1. While the manuscript describes interesting data, it is somewhat out of date, as the delta variant has become more dominant and has fueled the recent surge in new cases. If any comparison data between the alpha and delta variants are available, they should be included. It is recognized that the emergence of new variants is faster at this phase of the pandemic than is the ability to analyze their relative transmissibility, but such a comparison would enhance the manuscript.

Response: We acknowledge that additional data on the Delta VOC would enhance the manuscript. However, given the time needed to generate such comprehensive comparisons in vitro and in vivo, we believe that further delaying publication of our work to generate additional data with the Delta VOC may not be the best course of action. Instead, we have strengthened and optimized the manuscript cohesiveness and focus by performing additional studies, notably including (1) binding assays with Beta VOC spike, (2) in vitro competition with the Beta VOC in human nasal AECs, (3) in vivo competition with Beta in humanized ACE2-KI mice.

2. Page 7, para 2-It is curious that there is no fitness difference between the alpha variant and wt-S614G virus in hamsters but not in the other species that were analyzed. Differences might be more apparent if indirect transmission studies were added to the contact transmission studies.

Response: This is a good point and indirect transmission would be very interesting. However, all of our institutes are unfortunately not equipped to perform such studies under BSL3 conditions. We plan for such possibilities in the future.

Additional comments. 1. Both qRT-PCR and nextgen sequence analyses are described in the methods. Which one was used to generate the comparative sequence data shown in the figures?

Response: In this study, we have used qRT-PCR to generate the comparative analysis. We have clarified any ambiguous statements in the NGS methods section, where NGS was used exclusively to verify the sequence of the isolates and recombinant clones used in our studies. The NGS section was also moved to follow the “Virus” section in methods.

2. ED Figures 7,8, page 7, para 1-The beta variant becomes dominant in the lungs of some dually infected donor hamsters. At the same inoculum, is the beta variant more lethal than the alpha or S614G virus (this may be known from the literature already)?

Response: In our setup, we focused mainly on fitness advantages in terms of replication and transmission between Alpha, wt-S614G and Beta in a variety of animal models in a competitive approach and it was not our goal to compare the virulence/lethality of both individual variants, which would have necessitated a setup including e.g. single infections or dose titrations. The question about clinical dominance can therefore not be answered with our setup and should be part of additional studies, specifically designed to answer these questions. In the single virus infection in the context of the mild hACE2-KI model with Alpha, Beta, wt-S614G and wt-SAlpha, we did not observe more severe clinical signs with the Beta variant compared to Alpha or the ancestral wt-S614G. However, in the literature, there are controversial results concerning the severity of the disease associated with the Beta variant. Nevertheless, publications comparing the Alpha and Beta variant show that Syrian hamsters display more severe weight loss when infected with the Alpha variant compared to infection with Beta (ref.1,2). Moreover, histopathological lung scores in both of these studies were lower in Beta infected hamsters than in Alpha infected hamsters. This does indeed suggest a more severe and lethal disease for Alpha than for the Beta variant. (1) <https://www.biorxiv.org/content/10.1101/2021.03.11.435000v2.full.pdf> (Figure 1c) (2) <https://www.biorxiv.org/content/10.1101/2021.07.11.451964v1.full.pdf> (Figure S1B)

3. Page 9, para 2, Figure 5. The presence of encephalitis in the hACE2-K18Tg mice may reflect the use of young mice and a high dose of virus. This does not change the results but should be noted.

Response: Thank you. The development of encephalitis in hACE2-K18Tg associated with high dose infection is now acknowledged in the results section: “The increased replicative fitness of Alpha over wt-S614G was further reflected throughout the respiratory tract with higher genome copies in nose, lungs, olfactory bulb, and brain at 4 dpi (Fig. 5A), and all inoculated mice showed body weight loss at 4 dpi (Extended Data Fig. 11A). A relatively high infectious dose was used to promote transmission and was associated with high virus load (up to 10⁸ viral genome copies per sample) in the lung and brain leading to encephalitis as previously reported in hACE2-K18Tg mice 17,18.”

4. Page 9, 2 lines from bottom-Should be Fig. 5B.

Response: Thank you, the text has been corrected.

5. Page 10-first 4 lines-While the authors are properly cautious in interpreting the results, it seems equally likely that the results reflect small numbers of animals that were analyzed.

Response: In eight transmission pairs (half between wt-S614G and Alpha and half between wt-S614G and wt-SAlpha, we found 4 transmission events that involved only the Alpha variant or the isogenic clone expressing S Alpha; we were never able to detect wt-S614G in contact K18-Tg mice. We have acknowledged the limitations of the model for transmission as follows: “In the nose and

oropharyngeal swabs, viral loads were relatively lower and only limited transmission was observed (2 out of 4 contact). None of the contact mice lost weight, yet only Alpha was detectable in the lungs of contact mice 7 dpc (Extended Data Fig. 11).” And: “Interestingly, the replicative advantage of wt-SAlpha was less clear, where both wt-SAlpha and wt-S614G showed comparably high viral genome copies in lung and brain samples (Fig. 5B). Transmission to contact mice was inefficient, yet again wt-SAlpha was the only virus detected in lungs of contact mice 7 dpc (Extended Data Fig. 11).”

6. Page 10, 2 lines from bottom-should be Fig. 6E,F.

Response: Thank you, the text has been corrected accordingly

Author Response to Referee #2:

The authors have assembled a truly impressive amount of data in in vitro and in vivo models to assess the fitness of SARS-CoV-2 variants including the alpha and beta variants compared with a prototypic earlier wave virus. They employ both natural virus isolates and recombinant viruses engineered by reverse genetics to contain one or more Spike gene mutations present from the alpha VOC. Having one or more experimental models by which new and emerging variants could be risk assessed would be incredibly valuable for making public health decisions. Applying a reverse genetics approach to dissect which mutations in VOCs (that generally carry constellations of mutations) would also reveal valuable mechanistic information by which future variants can be assessed using sequence information alone, and by which we can better understand how this virus infects, causes disease and transmits. Overall the work is thus highly topical, of huge interest and importance.

However, the manuscript as presented currently lacks coherence. There is a lot of data, but none of it adds up to one single story and each part is somewhat incomplete. For example, the authors show that the human airway cultures do not reveal any differences between the replication of the alpha variant, or a recombinant virus bearing the alpha spike and the wild type precursor. Yet they go one to test replication of single point spike mutations in the airway cultures- it is not clear why they hypothesize they might detect differences in those mutants when the whole alpha variant isolate or spike recombinant did not show this. And indeed they find no differences between the mutants. On the other hand, they do show that in the ferret model and in the human ACE2 mouse there is a fitness advantage of the alpha variant- but in these models they do not test their set of single point recombinants. In other words, the recombinants have not really been put to use.

Similarly, they do show in the hamster model an advantage of alpha over beta but they do not test this in any other model, even not the in vitro cultures. They include a lot of data in supplementary figures that are not really essential to their story and duplicate work that has been performed and published extensively elsewhere by others such as neutralization data around alpha and beta VOCs.

On the other hand, they do perform some affinity assays using the ACE2 proteins from the various species they test, with wild type and alpha spike (but not beta) which could be really key in understanding why some animal models are more appropriate for measuring difference in VOCs than others, but this is a supp figure and hardly discussed, and the beta spike is not tested against hamster ACE2, the animal in which they did find a fitness difference.

Finally the whole presentation is rather haphazard with inconsistent use of log or non log viral titres, two different methods for neutralization assays, lack of inclusion of inoculum ratios for competition

assays in some but not all figures, and an array of various statistical analyses some of which do not seem appropriate (details below). To pull the data all together the authors conclude in this manuscript it is a good idea to use a lot of different model to assess variants. This is a useful point, but in this paper they do not cross compare with the models they have employed enough to really illustrate why one needs more models and which might be preferred, or why.

Response: We thank the reviewer very much for the words of appreciation and recognition of our work. The manuscript has been extensively revised to improve coherence and consistency in data presentation; and we are now able to provide a more decisive story line and a coherent presentation with key additional data comparing Beta in competition experiments in human nasal AEC cultures and humanized hACE2-KI mice. We have also notably streamlined the manuscript by removing the data on the individual Alpha spike point mutants and data on neutralization as indicated below in our specific responses.

Specific comments: Figure 1. illustrates in vitro work using primary cultures of human airway cells In A the authors employ nasal epithelia cells and measure viral RNA loads over time. There are only 2 replicates, and yet they plot mean and sd. Probably more appropriate to simply plot the data points. Are these data repeatable in cultures from different donors? Could more wells have been used?

Response: Cultures are comprised of nasal epithelial cells originating from 14 different donors. The individual data points are now shown.

In B they plot virus infectivity and only analyse the last time point by which time one can see from part A that exponential growth has finished. Differences might be much more pronounced at an earlier time point. These experiments should be repeated and more appropriate time points analysed by both RNA and infectious yield.

Response: We have now also included 4 hours post infection as requested.

The next set of in vitro experiment employs a competition assay in primary airway cells. Figure 1 C simply illustrates the genotype mix in the inoculum. This is not a real result and does not warrant its own figure, suggest incorporate C into part D.

Response: Agreed. This has now been integrated into a single panel figure 1C.

In part D we see the results of the competition experiments, B117 vs wt using two different B117 isolates, and also using a spike chimera It is not clear which statistical tests have been used here? The figure makes the wild type look fitter than B117?? Eg 120 hours lower left panel Are these really mean and SD? These are shown as box plots, And they are not symmetric (even taking into account the log scale)? Extended figure 1 shows virus replication kinetics with recombinant viruses harbouring single point mutations from the B117 spike. The presentation is really difficult to interpret; each virus is plotted alone -one can't really see if there are any difference, nor whether statistical analyses have been used to test whether there are any differences in replication?

Response: We have repeated the analysis and represented the data in a much simpler format showing individual data points. Additionally, three biological independent experiments have been performed using the Beta variant in the competition experiments, which are more stringent to detect increased fitness. We have performed a two-way ANOVA with the-Tukey Honest Significant

Differences post-hoc test. Extended Data Figure 1 including individual spike mutations has been removed from the manuscript as requested.

Extended figure 2 shows results of neutralization assays against alpha and beta vs wild type viruses with quite a limited set of sera. As stated above, this type of data has been shown already by multiple groups it is not novel at all... the authors could easily cite these other studies. Moreover the authors here have used two different assays, with quite different output titres, and it is not clear why. The data presentation is difficult, we can't see the individual results. What are the vertical lines in part A? Overall the data presentation in part B is easier to read.

Response: We agree that some supplementary data are not essential and replicate results performed by published studies. As a consequence, the neutralization assays have been removed from the revised manuscript and references to published work have been provided in the text.

Figures 2-3 and extended data figure 4 describes the results of Hamster competition assays which utilize a chain of direct contact transmission from a donor animal infected with a mix of viruses, Please show the inoculum ratios in the figure as done for the mouse experiments in figure 5. In the extended figures 7,8,9 the viral load in urt vs lrt are analysed; Extended figure 4 is not very clear cut.. there are clearly some contact animals with beta in lungs, the viral load titres written above each circle are quite difficult to unravel, might this data be better presented in a histogram? Is there really any reason to include extended figure 6 survival curves? for competition experiments how do you interpret these?

Response: We thank the reviewer for pointing out the missing inoculum ratios and we added these to all respective figures where it was absent. We acknowledge the reviewer's comment that the viral loads are difficult to unravel and the reviewer's suggestion to present the data in a histogram. However, the benefits of pie chart diagrams are the visualization of individual animals over time with virus ratios for each nasal washing / organ sample. In addition, the fixed assignment of individual animals to a specific cage is feasible by our illustration method. With the amount of individual data, a histogram figure would be far more massive and thus become extremely confusing. To integrate the viral loads in a more illustrative way, we sized each pie chart according to the level of viral RNA concentration. This gives a simple impression of viral shedding/ viral organ load at a glance, while exact values are presented in addition. Concerning the organ samples analyzed (Extended Figure 8), only Contact II from cage 4 showed a small fraction of Beta in the cranial lung. Only in donor animals, a substantial and predominant amount of Beta RNA was detected in the lung samples (Extended Fig7 and Fig8). All other contact animals only showed a near LOD amount of Beta genome in each organ sample analyzed. Moreover, the nasal washing samples completely match our findings within the nasal conchae samples of each respective animals. A similar outcome is observed in the experiment with wt-S614G vs Beta. Indeed, we agree with the reviewer's opinion that the survival curves do not add substantial information in a competitive transmission approach with varying individual starting times and inoculum doses. We therefore removed the figure from our manuscript as suggested.

In Figure 4 the authors turn to a different animal model: they use 6 sets of ferrets in the same style of competition experiment, a direct contact transmission chain, but this time only between wt vs b117 Infection and transmission in ferrets is much less efficient. Indeed the viral loads are very low. It is not clear whether sampling was stopped at day 6 for the contact 1 ferrets? Or whether all animals of 4 chains remained negative? Since one of the 2 actual transmissions was detected very late (day 5

after exposure) perhaps there were more contact 1 animals that became positive after day 6? The authors should make more clear in figure 4 what the grey circles demonstrate and which samples were collected. Nonetheless, the interesting conclusion is that in this highly stringent model one can detect fitness advantage of alpha VOC over wild type, this correlates with epidemiological experience and suggests the hamster model as used here not suitable for this type of fitness assessment going forwards. There is no discussion of whether using eg indirect contact or shorter exposure might make the hamster model more stringent and suitable.

Response: We thank the reviewer for appreciating the importance of our ferret study in the epidemiological context. We have now clarified the figure to show that the sampling and testing of all contact II ferrets was continued until the end of the experiment. We have improved legends to clarify that grey circles are used for samples with RT-qPCR value below the level of detection of the assay. We have acknowledged that indirect contact in the hamster model may be more stringent in the discussion of the “superspreader” hamster model. We agree that shorter exposure may be more stringent but it would be experimentally challenging to define appropriate conditions (time post infection, length of contact time) as they may differ between variants, which would require many more contact animals. We believe that using successive contact I and II hamsters makes the model more suitable and more compatible with reducing animal use. Indeed, we showed that in the competition between Alpha and wt-S614G, contact II animals are ultimately shedding principally only one of the two variants, however no clear transmission advantage was found as half of the contact II hamsters mainly shed Alpha and the other wt-S614G.

In figure 5 the authors then turn to the mouse models: first the K18 human ACE2 mice which also express mouse ACE2 and then a full humanized mouse where human ACE2 replaces murine ACE2 Like the ferret the mouse model is ‘inefficient’; and thus reveals the fitness advantage for B117 in line with human epidemiology. In part B the RG virus with B117 spike also shows an advantage over whole wild type virus, so here is a point at which perhaps the individual recombinants could have been tested? The presentation of raw genome data in these figure rather than log values is difficult for readers and inconsistent with earlier style. It is not clear exactly which criteria or statistical tests have been used to decide whether mice a ‘blue or orange’

Response: We agree with the reviewer’s comment that the hACE2-expressing mice would have been useful to test individual spike mutants. However, such experiments would have required a larger number of animals than the gain of knowledge to justify animal experimentation. We also agree that the use of the mutants in the nasal epithelial cells (formerly in Figure 1) are inconclusive since there was no difference between the S-Alpha and the S-614G in vitro. We have therefore decided to follow the recommendation and removed the description of the individual spike mutants from the revised manuscript. We apologize for the oversight concerning the genome load data presentation. The raw genome data below each pie chart in figure 5 and 6 are now presented identically as in figures 2, 3 and 4. To improve evaluation of the data, the size of each pie chart has been adjusted to correspond to the viral genome copies. We have also clarified the criteria for coloring the animal silhouettes in all the in vivo models: The color of corresponding VOC is attributed if there is a 2-fold or higher increase in genome copy number (>66%) in the last positive nasal wash (hamsters and ferrets) or oropharyngeal swabs (mice). This is now clearly stated in figure legends.

Extended figure 14 suggest some limited transmission in the mice : 2/4 transmissions, with always B117 predominant. These numbers are small, and the viral loads very low. How confident are the authors that this could be a feasible animal model going forwards?

Response: We agree that K18-hACE2 mice are not an efficient model to investigate transmission and we have therefore downplayed claims in the results section of the revised manuscript indicating the low efficiency. However, the model may remain interesting for the future, if a new VOC shows extraordinarily higher replication and/or specialization for the human ACE2. As indicated in the response to the other reviewer, in 4 out of 8 transmission pairs, wt-S614G was not found in the contact mice, only Alpha or wt-SAlpha was amplified.

Figure 6 use the humanized ACE2 mice and competes B117 vs wild type virus. Part D has high pfu in the lung for wt, but the genome copies in part C are the same as B117. The data done look entirely matched. Again, there are very low E gene copies and some below LOD. Is the mouse model a reliable model even though it does show up this difference?

Response: Different lung lobes were processed for RNA extraction and for plaque assays, which can explain discrepancies. We believe that the hACE2-KI mice are a reliable model with a replication kinetics that resembles that of most human infection events and similarly shows the fitness advantage of Alpha. Importantly, we have replicated the experiment with another group of 6 mice (females instead of males) with the same clear advantage for Alpha over wt-S614G (Extended Data Fig. 13A). The hACE2-expressing mouse models will also gain importance, if the virus is further adapting to the human receptor conditions, which might in the future generate differences to other animal models. In addition, we have now also compared Beta vs wt-S614G in this model. We could demonstrate that Beta is less fit than wt-S614G. Overall, the hACE2-KI studies match the results in the ferrets and provide additional evidence for fitness advantages of the Alpha VOC.

Finally there are a string of extended data figure around serological response of the infected animals for example supp figure 10 and 11. Here donor animals are tested- but we are told they were culled at day 4 or day 6- It doesn't seem very logical to test their antibody response? What was the hypothesis here? Ext fig 10 A doe contact 2 group to seroconvert? Ext fig 11 uses neutralization assays to analyses sera form animals after mixed infections. I think it is important here to understand wat the extent of cross reaction would be after single infections, otherwise such data is difficult to interpret.

Response: Thank you very much for your detailed analysis of the serological data. It is correct that we do not expect any SARS-CoV-2 specific serological response in the donor animals at this early time point. We basically see these values as confirmation of the serological negative status at the beginning of the experiment, as it is very difficult to obtain blood samples prior to inoculation from hamsters without euthanasia. In Extended Data Fig. 10A (now ED Fig.7) there is only one of the contact II ferrets positive in the ELISA. This confirms the PCR data showing only one contact II ferret becoming positive in the experiment. Indeed, we agree that the neutralization assays using animal sera resulting from mixed infections provide data that are difficult to interpret regarding crossreactivity and these data have been removed. These data however confirm an active productive infection giving rise to neutralizing antibodies and supporting the ELISA data this way.

Finally the concept of the hamster as a superspreader is a nice image, however in general superspreaders are individuals who for some reason transmit more efficiently eg due to higher bioaerosol exhalation- there is no measure of this here, rather that hamsters are highly susceptible to SARS-CoV-2. I am not sure the analogy quite works

Response: We agree that the idea of the hamster being for some issues more a “superspreader model” is a nice picture, but we also agree that there is restrictions concerning e.g. the strict aerosol spread, which was not tested here. We have therefore toned down this part in the discussion and referred primarily to replication and the very fast and efficient direct transmission. We also deleted the term from the headline at page 12 last paragraph.

Reviewer Reports on the First Revision:

Referee #2 (Response to First Revision):

The manuscript is significantly clearer and improved. I have no further comments or suggestions.